# Order-preserving pattern mining enhances structure-aware time series forecasting

## Abstract

Traditional time series forecasting models tend to focus on numerical fitting, making it difficult to explicitly model and leverage the relative ordering patterns inherent in time series. This often results in suboptimal predictions when dealing with data segments that exhibit clear pattern regularities. To address this gap, this paper introduces Order-Preserving Patterns (OPPs) into time series forecasting for the first time and proposes a novel model that explicitly incorporates prior pattern knowledge by leveraging frequent OPPs as explicit priors. The proposed model utilizes a convolutional neural network to perform feature dimensionality reduction on high-dimensional labeled time series, extracting one-dimensional representations suitable for pattern mining. It then applies a sliding window and support counting strategy to discover frequent OPPs. An OPP matching mechanism is proposed to distinguish between OPP and non-OPP training samples. Additionally, a pattern constrained loss function is designed to guide the predicted values toward consistency with the prior pattern logic. This constraint is imposed from three perspectives—right boundary, left boundary, and intermediate positions—to ensure order alignment with the tail elements of the OPPs. Experimental results show that under the 'Perturbation Boundary' window sizes across ten real-world and public benchmark datasets, the proposed OPPCL model consistently achieves substantially lower MSE compared with state-of-the-art methods. In particular, it yields at least 31.45% and 37.30% reductions on the SWaT and Electricity datasets, respectively. The improvement becomes more pronounced when the window size exceeds the 'Perturbation Boundary'. Code is available at this repository: `https://anonymous.4open.science/r/OPPCL-B070/`.

## 1 Introduction

With the rapid development of the Industrial Internet of Things (IIoT) and edge sensing technologies, industrial control systems can now collect key equipment states, environmental parameters, and operational indicators in real time, generating massive time series data across domains such as smart manufacturing, power monitoring, chemical processes, traffic scheduling, and industrial safety (Sun et al., 2021; Tu et al., 2025). In these applications, forecasting models are widely used to predict future states, detect trend changes, and identify potential risks, playing a critical role in system regulation and intelligent decision-making (Zhang et al., 2024; Wang et al., 2022). Consequently, building high-precision forecasting models based on historical time series has become a core task in IIoT-driven intelligent modeling scenarios (Uchiteleva et al., 2022; Nie et al., 2021). In recent years, fueled by the progress of deep learning, models based on Recurrent Neural Networks (RNNs) (Kacprzyk et al., 2024), Convolutional Neural Networks (CNNs) (Dai et al., 2024), and Transformer architectures (Wu et al., 2020; Panigrahi et al., 2024) have been widely adopted for time series forecasting tasks. These models have achieved promising results on various open-source datasets, with Transformers and their variants particularly excelling in medium- and long-term forecasting tasks due to their ability to model long-range dependencies. However, existing forecasting models generally lack order pattern awareness and pattern-level supervision, making it difficult to capture and preserve the stable logical relations in industrial control data, which leads to structurally inconsistent predictions and degraded stability and reliability, as illustrated in Figure 1.

To address the aforementioned challenges, this paper proposes a novel time series forecasting model with OPP awareness, termed OPPCL (Order-Preserving Pattern Constrained Learning model). The

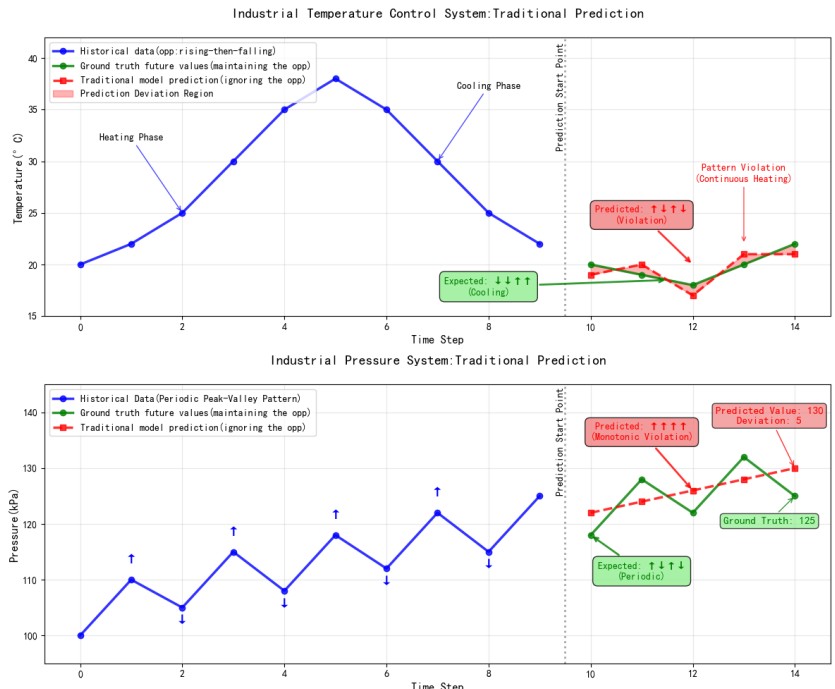

Figure 1: Impact of Ignoring OPPs on Prediction Consistency in Industrial Time Series.

model incorporates frequent OPPs as structural priors and establishes a supervision mechanism based on relative order, thereby enhancing its predictive accuracy and structural consistency on pattern-rich sequences. The main contributions of this work are as follows:

1) We propose a pattern-aware forecasting model, OPPCL, which for the first time explicitly introduces frequent OPPs as structural priors in the time series forecasting task. This breaks the limitations of traditional models that lack the ability to model and utilize sequential structural patterns.

2) We design a position-sensitive pattern-constrained loss function, which leverages the intrinsic ordering logic of frequent OPPs to construct boundary-based ranking constraints. This loss term explicitly supervises the relative order of the model outputs, guiding the model to produce structurally consistent predictions in pattern-dense regions.

3) We develop a complete OPP-aware forecasting framework comprising: (1) a dimensionality reduction module; (2) a frequent OPP mining module based on sliding windows and support counting; (3) a pattern matching mechanism for dynamic sample differentiation; and (4) a structure-constrained optimization process integrating relative order-based supervision with value regression.

4) We conduct empirical evaluations on ten real-world datasets. The results demonstrate that OPPCL consistently outperforms mainstream baseline methods in terms of prediction accuracy on frequent OPP samples.

## 2 RELATED WORK

Time series forecasting plays a vital role in domains such as industrial operation, energy scheduling, and traffic control (Yang et al., 2022; Luo et al., 2025). Classical statistical models, including AR, MA, and ARIMA (Bergamin et al., 2024; Koolen et al., 2015; Miao et al., 2023), are effective for linear and stationary data but underperform on nonlinear, noisy, or long-range dependent sequences. Deep learning models such as RNNs, LSTMs, GRUs (Yao et al., 2023; Dridi et al., 2022; Song et al., 2021), and Transformer-based architectures including Informer and Autoformer (Kong et al.,

2025; Xu et al., 2023; Wu et al., 2021), have demonstrated strong capability in capturing temporal dependencies. However, these methods primarily rely on pointwise regression objectives, making them insensitive to internal structural patterns such as relative ordering or pattern consistency (Liu et al., 2024). This often results in unstable predictions when data exhibit strong local regularities.

Recent works have attempted to incorporate structural priors into time series models. CNN–RNN hybrids such as LSTNet (Wang & Chen) and TPA-LSTM (Chen et al., 2023) integrate convolutional encoders to capture short-term local patterns, while models like DeepGLO and TimesNet (Sen et al., 2019; Wu et al., 2023a) introduce frequency-domain decomposition to enhance periodic structure modeling. More recently, multivariate structure modeling has been revitalized through learnable decompositions and inter-series dependency modeling (Yu et al., 2024), showing that explicitly modeling structural variations significantly benefits forecasting. Nevertheless, these approaches generally learn structural information implicitly, without externally guided or interpretable structure priors.

In parallel, a growing body of work investigates shape-based or pattern-based representations for time series. Shapelet-based learning has evolved from supervised discriminative shapelets to unsupervised general shapelets through contrastive learning, as demonstrated in TimeCSL (Liang et al., 2024). Germain T also introduced a general framework for shape-based analysis that focuses on subsequence geometry and shape similarity (Germain et al., 2024). These methods highlight the importance of local structural motifs but mainly target shape geometry rather than value ordering. In contrast, ordered-pair patterns (OPPs) emphasize the relative ranking relationships within subsequences (Wu et al., 2024; Li et al., 2024), e.g., "increase → decrease". Algorithms such as OPR-Miner (Wu et al., 2023b) efficiently discover frequent OPPs, but their integration into forecasting models remains unexplored.

Loss functions guide model optimization, but commonly used pointwise losses such as MSE and MAE (Dina et al., 2023; Liu et al., 2022) fail to capture relative structures and ordering logic in time series. While ranking losses (Oksuz et al., 2021) and local consistency constraints (Yu et al., 2021) have improved structural coherence in fields like vision and NLP, they are rarely applied to time series. Some works consider shape or period-alignment losses (Le Guen & Thome, 2023a) and stage-penalization mechanisms (Zheng et al., 2025), yet these are limited to periodic data or rely on external labels. Le Guen proposed prediction criteria that incorporate shape- and time-aware structures (Le Guen & Thome, 2023b), which explicitly enforce shape similarity and temporal consistency in forecasting. However, such approaches typically focus on geometric and phase-level alignment, rather than imposing constraints on relative-order–based patterns. Furthermore, most models use a uniform loss across all samples, restricting their ability to exploit structural priors in highly patterned sequences.

In summary, existing studies either: (1) learn structural patterns implicitly (e.g., shapelets, decompositions), (2) focus on geometric shape similarity rather than ordering, or (3) lack mechanisms to enforce interpretable structural constraints.

To bridge this gap, we propose OPPCL, the first forecasting model that explicitly incorporates frequent OPPs as structure-aware supervisory signals, enabling models to enforce relative-ordering regularities during optimization.

## 3 PROPOSED METHOD

To enable the model to fully utilize the mined OPPs as prior knowledge during training, thereby improving its prediction accuracy on pattern-conforming samples, we propose a pattern constrained loss function and explore a model architecture named OPPCL tailored to this loss. The overall architecture is illustrated in Figure 2. This section provides a detailed description of its working mechanism and theoretical foundations. The dimensionality reduction process designed for labeled time series data is described in Section 3.1. The procedure for mining prior knowledge from the reduced data is presented in Section 3.2. To distinguish between frequent OPP samples and non-OPP samples during training, we design a pattern matching detection mechanism, as detailed in Section 3.3. Finally, we present the training pipeline for OPPCL in Section 3.4, where the proposed pattern constrained loss function serves as the core component.

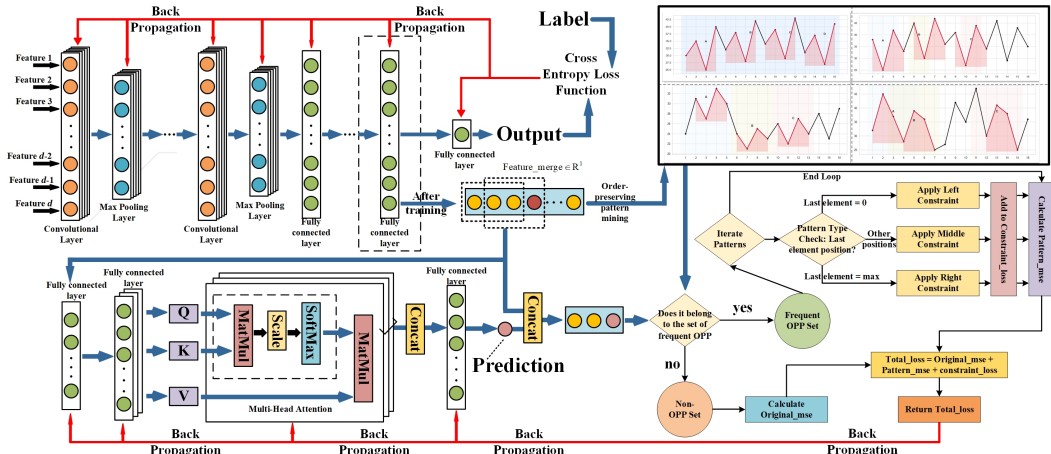

Figure 2: The overall architecture of OPPCL. The top-left section illustrates the dimensionality reduction module designed for labeled time series data. The top-right section presents several frequent OPPs extracted from the reduced data. The bottom-left section depicts the main structure of the OPPCL model, which consists of an encoder module based on attention mechanisms and fully connected layers. The bottom-right section highlights the core of OPPCL—namely, the pattern matching mechanism and the pattern constrained loss function.

## 3.1 DATA DIMENSIONALITY REDUCTION PROCESS

Given a time series $S = (\mathbf{s}_1, \mathbf{s}_2, \ldots, \mathbf{s}_n)$, where $n$ denotes the number of time steps and each $\mathbf{s}_i \in \mathbb{R}^d$ represents a feature vector at time step $i$ with dimensionality $d$, the entire time series can be represented in matrix form as $S \in \mathbb{R}^{n \times d}$.

OPPCL employs a CNN to learn feature representations from the time series. Let the parameters of the CNN be denoted by $\theta_{\text{CNN}}$, and the output of the network be $S' = f_{\theta_{\text{CNN}}}(S) \in \mathbb{R}^{n \times c}$, where $c$ is the number of classes, and $f_{\theta_{\text{CNN}}}$ denotes the mapping function during training. The objective of training the CNN within OPPCL is to minimize the loss function as defined in (1):

$$\mathcal{L}(\theta_{\text{CNN}}) = \sum_{i=1}^{n} \mathcal{L}\big(f_{\theta_{\text{CNN}}}(\mathbf{s}_i), y_i\big) \tag{1}$$

where $y_i$ denotes the class label corresponding to $\mathbf{s}_i$, and $\mathcal{L}(\cdot)$ represents the cross-entropy loss function.

After fully training the CNN as described above, the parameters are fixed and denoted by $\theta_{\text{CNN}}^*$. A switch is inserted at the penultimate layer of the CNN to extract the intermediate features, enabling dimensionality reduction. The intermediate output of the CNN is controlled to satisfy (2):

$$\mathbf{S}_{\text{inter}} = g_{\theta_{\text{CNN}}^*}(S) \in \mathbb{R}^{n \times 1} \tag{2}$$

where $g_{\theta_{\text{CNN}}^*}$ denotes the dimensionality reduction mapping function based on the trained CNN.

## 3.2 OPP-MINING FOR PRIOR KNOWLEDGE EXTRACTION

After the dimensionality reduction process described in Section 3.1, the original time series $S = (\mathbf{s}_1, \mathbf{s}_2, \ldots, \mathbf{s}_n)$ is transformed into a one-dimensional time series $\mathbf{S}_{\text{inter}} = (s_{\text{inter}_1}, s_{\text{inter}_2}, \ldots, s_{\text{inter}_n}) \in \mathbb{R}^{n \times 1}$.

Given a minimum support threshold $minsup$, an expected sliding window size $v$, and a sliding step size $step$, we perform the following operations. For each pattern length $l$ from 2 to $v$, time series $\mathbf{S}_{\text{inter}}$ is partitioned into a set of sub-time series using a window of size $l$ and a sliding step of size $step$, as shown in (3):

$$\mathcal{W} = \Big\{ \mathbf{S}^{(i)} = (s_{\text{inter}_i}, s_{\text{inter}_{i+1}}, \ldots, s_{\text{inter}_{i+l-1}}) \, \Big| \, 1 \le i \le n - l + 1, \, i = 1 \pmod{step} \Big\} \tag{3}$$

For each sub-time series $\mathbf{S}^{(i)} \in \mathcal{W}$, its OPP is denoted by $R(\mathbf{S}^{(i)}) \in \mathcal{S}_v$ (as defined in Definition 2), where $\mathcal{S}_v$ represents the set of all permutations of length $v$. Let $\mathbf{p} \in \mathcal{S}_l$ be an OPP with length $l$. The support of $\mathbf{p}$ is defined as the number of sub-time series in which $\mathbf{p}$ appears:

$$supp(\mathbf{p}) = \left| \left\{ \mathbf{S}_{[l]}^{(i)} \in \mathcal{W}_l \mid R(\mathbf{S}_{[l]}^{(i)}) = \mathbf{p} \right\} \right| \tag{4}$$

where $\mathcal{W}_l \subseteq \mathcal{W}$ denotes the set of all sliding window sub-time series of length $l$.

The frequent OPPs set of length $l$ is defined as $F_l = \{ \mathbf{p} \in \mathcal{S}_l \mid sup(\mathbf{p}, \mathbf{S}_{\text{inter}}) \geq minsup \}$. Starting with $l = 2$, all length-2 frequent OPPs $F_2$ are computed. If two patterns $\mathbf{p} \in F_l$ and $\mathbf{q} \in F_l$ satisfy condition $R(suffix(\mathbf{p})) = R(prefix(\mathbf{q}))$, then a new super-pattern $\mathbf{r}$ (or $\mathbf{r}$ and $\mathbf{h}$) of length $l + 1$ can be generated according to Definition 6. For each generated super-pattern, we compute its support. If the support satisfies $sup(\mathbf{r}, \mathbf{S}_{\text{inter}}) \geq minsup$, then the pattern is added to $F_{l+1}$. This process is repeated until the pattern length reaches $v$.

Finally, the set of frequent OPPs with length $v$ and their corresponding support values are obtained, as shown in (5):

$$F_v = \{ \mathbf{p}_v \in \mathcal{S}_v \mid sup(\mathbf{p}_v, \mathbf{S}_{\text{inter}}) \geq minsup \} \tag{5}$$

where each $\mathbf{p}_v$ denotes an OPP with length $v$, and the corresponding support values are stored in array $\mathbf{svae} = [\, sv_1, sv_2, \ldots, sv_{|F_v|} \,]$.

## 3.3 Pattern matching detection method

Given a sequence segment $\mathbf{X} = (s_{\text{inter}_i}, s_{\text{inter}_{i+1}}, \ldots, s_{\text{inter}_{i+v-2}})$, it is used as input to the OPPCL model. Each scalar $s_{\text{inter}_j}$ in the input segment is embedded into $\mathbf{z}_j$, yielding $Z = [\mathbf{z}_1, \ldots, \mathbf{z}_{v-1}]^\top$. The Transformer encoder produces the final representation $\mathbf{z}'_{v-1}$, and the output is $\hat{y} = W_o^\top \mathbf{z}'_{v-1} + b$.

Next, OPPCL concatenates the original time series segment $\mathbf{X}$ and predicted value $\hat{y}$ to construct a new time series with length $v$, denoted by $\hat{\mathbf{X}} = [s_{\text{inter}_i}, s_{\text{inter}_{i+1}}, \ldots, s_{\text{inter}_{i+v-2}}, \hat{y}]$.

According to Definition 2, the model obtains OPP $\mathbf{X}' = R(\hat{\mathbf{X}})$, and then determines whether $\mathbf{X}'$ belongs to the set of frequent OPPs $F_v$. This pattern-matching mechanism is formally defined as:

$$\begin{cases} \mathcal{D}_{\text{pattern}} = \{(\mathbf{X}, \hat{y}) \mid R(\hat{\mathbf{X}}) \in F_v\} \\ \mathcal{D}_{\text{non-pattern}} = \{(\mathbf{X}, \hat{y}) \mid R(\hat{\mathbf{X}}) \notin F_v\} \end{cases} \tag{6}$$

Here, $\mathcal{D}_{\text{pattern}}$ represents the set of samples forming frequent OPPs of length $v$, while $\mathcal{D}_{\text{non-pattern}}$ denotes the set of samples forming non-OPPs with the same length.

## 3.4 Loss function of OPPCL

To further enhance the consistency between the model-generated time series and the OPPs mined from real data, OPPCL introduces a structural **pattern constraint loss** on top of the standard MSE loss. This constraint acts as a component of the overall loss function and explicitly encourages the model to learn and generate predictions that conform to frequent OPP rules during training. The total objective loss function comprises three core components: basic prediction loss, pattern-region prediction loss, and pattern constraint loss, and is formally defined as:

$$\mathcal{L}_{\text{total}} = \lambda_1 \mathcal{L}_{\text{basic}} + \lambda_2 \mathcal{L}_{\text{pattern}} + \lambda_3 \mathcal{L}_{\text{constraint}} \tag{7}$$

where $\lambda_1$, $\lambda_2$, and $\lambda_3$ are hyperparameters that control the relative importance of each loss component.

### 3.4.1 Basic prediction loss

The basic prediction loss is defined over the sample set $\mathcal{D}_{\text{non-pattern}}$, which consists of length-$v$ time series that are non-OPP. It is formulated as:

$$\mathcal{L}_{\text{basic}} = \frac{1}{|\mathcal{D}_{\text{non-pattern}}|} \sum_{(\mathbf{X}, \hat{y}) \in \mathcal{D}_{\text{non-pattern}}} (\hat{y} - y)^2 \tag{8}$$

This component measures the model's fundamental predictive performance on non-OPP samples without enforcing any structural constraints.

### 3.4.2 PATTERN-REGION PREDICTION LOSS

The pattern-region prediction loss is defined over the sample set $\mathcal{D}_{\text{pattern}}$, which contains time series with length $v$ that correspond to frequent OPPs. It is given by:

$$\mathcal{L}_{\text{pattern}} = \frac{1}{|\mathcal{D}_{\text{pattern}}|} \sum_{(\mathbf{X},\hat{y})\in\mathcal{D}_{\text{pattern}}} (\hat{y} - y)^2 \tag{9}$$

Unlike basic loss $\mathcal{L}_{\text{basic}}$, the pattern-region prediction loss focuses on samples with structural pattern characteristics, thereby preventing unstructured prediction signals from interfering with the training of structurally-aware predictions.

### 3.4.3 PATTERN CONSTRAINT LOSS

According to Section 3.2, the total support of the frequent OPPs set $F_v$ is calculated as:

$$st = \sum_{i=1}^{|F_v|} sv_i \tag{10}$$

To reflect the relative importance of each pattern, all support values are normalized into a probability distribution array within the range $[0, 1]$, denoted by:

$$\mathbf{slist} = \left[ \frac{sv_1}{st}, \frac{sv_2}{st}, \ldots, \frac{sv_{|F_v|}}{st} \right] \tag{11}$$

As defined in Section 3.3, let the input time series with predicted value be $\hat{\mathbf{X}} = [s_{\text{inter}_i}, s_{\text{inter}_{i+1}}, \ldots, s_{\text{inter}_{i+v-2}}, \hat{y}]$. Suppose the $i$-th frequent OPP in $F_v$ is $F_v^i$. During the constraint process in OPPCL (looping from 1 to $|F_v|$), there are three cases:

**Case 1:** $F_v^i[-1] = 0$. In this case, position 1 in $\hat{\mathbf{X}}$ corresponds to a reference value $ref_r$, which is expected to be greater than $\hat{y}$ in order to comply with the OPP. Therefore, the constraint logic is $\hat{y} < ref_r - \epsilon$. The corresponding right-boundary constraint loss is defined as:

$$\mathcal{L}_{\text{constraint}}^r = \frac{1}{|\mathcal{D}_{\text{pattern}}|} \sum_{(\mathbf{X},\hat{y})\in\mathcal{D}_{\text{pattern}}} \frac{sv_i}{st} \cdot \max(0, \hat{y} - ref_r + \epsilon) \tag{12}$$

**Case 2:** $F_v^i[-1] = v - 1$. In this case, position $v - 2$ in $\hat{\mathbf{X}}$ corresponds to a reference value $ref_l$, which is expected to be smaller than $\hat{y}$. Therefore, the constraint logic is $\hat{y} > ref_l + \epsilon$. The left-boundary constraint loss is defined as:

$$\mathcal{L}_{\text{constraint}}^l = \frac{1}{|\mathcal{D}_{\text{pattern}}|} \sum_{(\mathbf{X},\hat{y})\in\mathcal{D}_{\text{pattern}}} \frac{sv_i}{st} \cdot \max(0, ref_l - \hat{y} + \epsilon) \tag{13}$$

**Case 3:** $0 < F_v^i[-1] < v - 1$. In this case, the reference indices in $\hat{\mathbf{X}}$ are $g_{left} = \mathbf{F}_v^i[-1] - 1$ and $g_{right} = F_v^i[-1] + 1$. The value at $g_{left}$, denoted by $val_l$, is expected to be smaller than $\hat{y}$, and the value at $g_{right}$, denoted by $val_r$, is expected to be greater than $\hat{y}$. The constraint logic is $val_l + \epsilon < \hat{y} < val_r - \epsilon$. The corresponding middle-boundary constraint loss is defined as:

$$\mathcal{L}_{\text{constraint}}^m = \frac{1}{|\mathcal{D}_{\text{pattern}}|} \sum_{(\mathbf{X},\hat{y})\in\mathcal{D}_{\text{pattern}}} \left( \frac{sv_i}{st} \times \left(\max\left(0, val_l - \hat{y} + \epsilon\right) + \max\left(0, \hat{y} - val_r + \epsilon\right)\right) \right) \tag{14}$$

At the end of each iteration, OPPCL accumulates the corresponding constraint loss into the total pattern constraint loss:

$$\mathcal{L}_{\text{constraint}} = \sum_{i=1}^{|F_v|} \mathcal{L}_{\text{constraint}}^{(i)} \tag{15}$$

where

$$\mathcal{L}_{\text{constraint}}^{(i)} = \begin{cases} \mathcal{L}_{\text{constraint}}^r, & \text{if } F_v^i[-1] = 0 \\ \mathcal{L}_{\text{constraint}}^l, & \text{if } F_v^i[-1] = v - 1 \\ \mathcal{L}_{\text{constraint}}^m, & \text{if } 0 < F_v^i[-1] < v - 1 \end{cases}$$

## 4 EXPERIMENTS

### 4.1 PROTOCOLS

**Datasets** Ten datasets are used to evaluate OPPCL: SWaT dataset, ETTh1 dataset, ETTh2 dataset, ETTm1 dataset, ETTm2 dataset, PowerSystem dataset, Weather dataset, Electricity dataset, Exchange_Rate dataset, Traffic dataset. The statistics of the datasets and the detailed dimensionality reduction procedures of OPPCL for the two datasets are presented in Table 3(Appendix C.1). The specific model architecture parameters and training parameter settings of OPPCL are presented in Table 4(Appendix C.3).

**Baseline** We compare OPPCL with ten comparison models: Autoformer (Wu et al., 2021), ConvLSTM (Lin et al., 2020), Crossformer (Zhang & Yan, 2023), Informer (Xu et al., 2023), LSTNet (Li et al., 2020), MegaCRN (Jiang et al., 2023), SCINet (LIU et al., 2022), and Leddam (Yu et al., 2024), TimeCSL(Liang et al., 2024), CycleNet(Lin et al., 2024). All comparison models are configured using their default settings.

**Ablation Study** To evaluate the performance of pattern constraint loss, we conduct an ablation study between two models: the full OPPCL model with the order-preserving constraint loss and its counterpart w/o CL, which removes the pattern constraint loss. The purpose of this comparison is to demonstrate the critical role of the constraint mechanism. Aside from the inclusion or exclusion of the constraint loss, the two models share identical parameter settings.The evaluation metrics for the above models (including the baseline models) are presented in Appendix C.2.

### 4.2 OPP-MINING

We apply OPP-Mining (Wu et al., 2023b) to extract frequent OPPs from ten datasets. The related parameters are shown in Table 4(Appendix C.3). For example, From the SWaT dataset, we mine frequent OPPs with lengths ranging from 2 to 13. From the ETTh1 dataset, we obtain patterns with lengths ranging from 2 to 9. Due to the large number of extracted patterns, we only report patterns with length $\geq 8$ for SWaT and length $\geq 6$ for ETTh1, and these results are summarized in Appendix C.4.

### 4.3 MAIN RESULTS

The primary MSE results are presented in Table 1, and the MAE results are reported in Table 2. Table 1 compares the MSE losses of different models across ten datasets under varying sliding window sizes, where the pattern length corresponds to the window size used in the experiments. For the SWaT dataset, when the sliding window size is greater than or equal to 9, OPPCL demonstrates superior predictive accuracy, with the mean squared error at least one order of magnitude lower than that of the baseline models. When the pattern length is 9, the MSE reaches as low as $9.740\mathrm{e}{-4}$, representing a 31.45% reduction compared with the current SOTA models. Similarly, for the remaining datasets, OPPCL also achieves strong performance once the window size exceeds a certain boundary—such as 9 for SWaT, 7 for ETTh1, 6 for PowerSystem, and 6 for Weather. Table 2 shows a similar trend. For example, on the SWaT dataset, OPPCL achieves at least a 31.46% reduction in MAE compared with the SOTA models when the window size is 9. On the PowerSystem dataset, OPPCL attains at least a 63.2% MAE reduction when the window size is 6. This phenomenon can be explained as follows: as shown in Tables 5 and 6 of Appendix C.5 (we present the OPP mining results for two of the datasets, SWaT and ETTh1), the number of frequent OPPs mined from the SWaT dataset with lengths $\geq 9$ is significantly smaller than that of patterns with lengths $< 9$. When too many frequent OPPs are mined, it can lead to a certain degree of 'disturbance' in the constraint loss of OPPCL, making it harder for the model to balance the abstraction among various patterns during training, we provide an analysis of this issue in Appendix C.4. The same issue arises in the ETTh1 dataset, where frequent OPPs of length $< 7$ are more abundant, resulting in a similar 'prior knowledge imbalance' problem. We refer to this boundary of pattern length as the 'Perturbation Boundary', abbreviated as 'PB' in the tables.

In addition, we evaluated the long-term forecasting capability of OPPCL. Theoretically, OPPCL specifies a window size based on the mined pattern lengths: all elements in the window except the last one are used as pattern-matching signals, while the prediction of the final element is structurally

constrained (see Section 3.4.3 for details). Consequently, the maximum feasible window length is determined by the length of the most frequent OPP obtained from prior knowledge. To conduct long-term forecasting while preserving the structural constraints, we apply the following strategy during testing: OPPCL performs recursive prediction, where at each step the most recent $v-1$ real or previously predicted values are fed into the model to predict the next point $\hat{y}_{t+1}$. The prediction is then appended to the sequence, and the process is rolled forward until the desired horizon is reached. The long-term forecasting results of OPPCL on the SWaT and ETTh1 datasets are presented in Appendix C.10.

Table 1: MSE Comparison Across Models on Multiple Datasets.

| Dataset | Horizon | Autoformer | ConvLSTM | Crossformer | Informer | LSTNet | MegaCRN | SCINet | Leddam | TimeCSL | CycleNet | w/o CL | OPPCL |
|---|---|---|---|---|---|---|---|---|---|---|---|---|---|
| SWaT | 7 | 1.495e-3 | 1.478e-3 | 3.308e-3 | 1.874e-3 | **1.431e-3** | 1.451e-3 | 2.485e-2 | 1.664e-3 | 1.511e-3 | 1.443e-3 | 2.699e-3 | 2.858e-3 |
| | 8 | 1.655e-3 | 1.535e-3 | 1.512e-3 | 2.245e-3 | **1.432e-3** | 1.450e-3 | 2.552e-2 | 1.655e-3 | 1.514e-3 | 1.457e-3 | 2.861e-3 | 2.887e-3 |
| | **9(PB)** | 1.576e-3 | 1.513e-3 | 1.841e-2 | 1.170e-2 | 1.421e-3 | 1.448e-3 | 9.653e-3 | 1.860e-3 | 1.476e-3 | 1.450e-3 | 3.078e-3 | **9.740e-4** |
| | 10 | 1.681e-3 | 1.527e-3 | 1.493e-3 | 2.845e-3 | 1.417e-3 | 1.438e-3 | 1.255e-1 | 1.666e-3 | 1.488e-3 | 1.438e-3 | 3.405e-3 | **9.870e-4** |
| | 11 | 1.583e-3 | 1.433e-3 | 3.414e-3 | 2.202e-3 | 1.420e-3 | 1.426e-3 | 4.198e-2 | 1.558e-3 | 1.486e-3 | 1.451e-3 | 3.371e-3 | **4.240e-4** |
| | 12 | 1.660e-3 | 1.615e-3 | 2.111e-3 | 1.493e-3 | 1.423e-3 | 1.481e-3 | 9.874e-2 | 1.590e-3 | 1.498e-3 | 1.453e-3 | 3.447e-3 | **1.000e-6** |
| | 13 | 1.519e-3 | 1.543e-3 | 1.643e-2 | 1.114e-2 | 1.417e-3 | 1.419e-3 | 1.168e-1 | 1.585e-3 | 1.591e-3 | 1.472e-3 | 3.649e-3 | **1.400e-4** |
| ETTh1 | 6 | 4.322e-1 | 4.296e-1 | 4.216e-1 | 4.229e-1 | 4.263e-1 | 4.325e-1 | **4.149e-1** | 4.246e-1 | 4.237e-1 | 5.070e-1 | 4.449e-1 | 7.094e-1 |
| | **7(PB)** | 4.375e-1 | 4.299e-1 | 4.198e-1 | 4.265e-1 | 4.249e-1 | 4.330e-1 | 4.116e-1 | 4.264e-1 | 4.309e-1 | 5.310e-1 | 4.495e-1 | **3.937e-1** |
| | 8 | 4.414e-1 | 4.304e-1 | 4.237e-1 | 4.287e-1 | 4.257e-1 | 4.335e-1 | 4.126e-1 | 4.285e-1 | 4.345e-1 | 5.251e-1 | 4.558e-1 | **3.093e-1** |
| | 9 | 4.456e-1 | 4.307e-1 | 4.230e-1 | 4.319e-1 | 4.276e-1 | 4.339e-1 | 4.125e-1 | 4.294e-1 | 4.310e-1 | 5.912e-1 | 4.601e-1 | **5.412e-2** |
| ETTh2 | 6 | 2.013e-1 | 2.102e-1 | **1.742e-1** | 1.764e-1 | 1.839e-1 | 1.965e-1 | 2.109e-1 | 1.749e-1 | 1.831e-1 | 2.102e-1 | 2.227e-1 | 2.248e-1 |
| | **7(PB)** | 2.018e-1 | 2.117e-1 | 1.742e-1 | 1.779e-1 | 1.826e-1 | 1.967e-1 | 1.903e-1 | 1.747e-1 | 1.805e-1 | 2.135e-1 | 2.432e-1 | **1.719e-1** |
| | 8 | 2.028e-1 | 2.133e-1 | 1.752e-1 | 1.749e-1 | 1.826e-1 | 1.969e-1 | 2.044e-1 | 1.762e-1 | 1.794e-1 | 2.160e-1 | 2.503e-1 | **4.749e-2** |
| ETTm1 | 6 | 1.404e-1 | 1.351e-1 | 1.348e-1 | 1.338e-1 | 1.323e-1 | 1.357e-1 | 1.492e-1 | 1.329e-1 | 1.328e-1 | **1.311e-1** | 1.474e-1 | 2.043e-1 |
| | **7(PB)** | 1.414e-1 | 1.351e-1 | 1.348e-1 | 1.347e-1 | 1.324e-1 | 1.357e-1 | 1.486e-1 | 1.330e-1 | 1.330e-1 | 1.322e-1 | 1.476e-1 | **4.959e-2** |
| | 8 | 1.413e-1 | 1.351e-1 | 1.369e-1 | 1.330e-1 | 1.325e-1 | 1.357e-1 | 1.368e-1 | 1.329e-1 | 1.325e-1 | 1.326e-1 | 1.512e-1 | **2.001e-2** |
| | 9 | 1.421e-1 | 1.351e-1 | 1.354e-1 | 1.335e-1 | 1.327e-1 | 1.357e-1 | 1.425e-1 | 1.328e-1 | 1.324e-1 | 1.331e-1 | 1.501e-1 | **2.270e-4** |
| ETTm2 | 6 | 7.949e-2 | 7.824e-2 | 7.557e-2 | **7.423e-2** | 7.715e-2 | 9.071e-2 | 1.753e-1 | 7.569e-2 | 8.595e-2 | 7.479e-2 | 1.076e-1 | 1.999e-1 |
| | 7 | 7.845e-2 | 7.824e-2 | 7.538e-2 | 7.935e-2 | 7.633e-2 | 8.924e-2 | 1.295e-1 | 7.708e-2 | 8.239e-2 | **7.472e-2** | 1.154e-1 | 1.161e-1 |
| | **8(PB)** | 7.724e-2 | 7.823e-2 | 7.493e-2 | 7.729e-2 | 7.637e-2 | 8.642e-2 | 2.009e-1 | 8.675e-2 | 8.113e-2 | 7.512e-2 | 1.228e-1 | **4.043e-2** |
| | 9 | 7.690e-2 | 7.820e-2 | 7.836e-2 | 7.783e-2 | 7.639e-2 | 8.578e-2 | 1.306e-1 | 7.698e-2 | 7.744e-2 | 7.659e-2 | 1.250e-1 | **6.200e-5** |
| PowerSystem | 4 | 9.625e-1 | 9.508e-1 | 9.675e-1 | 9.732e-1 | 9.948e-1 | 1.045e-0 | 9.993e-1 | 9.742e-1 | 9.425e-1 | **9.176e-1** | 1.050e-0 | 1.088e-0 |
| | 5 | 9.665e-1 | 9.509e-1 | 9.746e-1 | 9.635e-1 | 1.001e-0 | 1.043e-0 | 1.044e-0 | 9.746e-1 | 9.294e-1 | **9.161e-1** | 1.103e-0 | 8.960e-1 |
| | **6(PB)** | 9.664e-1 | 9.507e-1 | 9.770e-1 | 9.616e-1 | 1.021e-0 | 1.042e-0 | 1.020e-0 | 9.671e-1 | 9.127e-1 | 9.211e-1 | 1.152e-0 | **3.358e-1** |
| | 7 | 9.651e-1 | 9.501e-1 | 9.765e-1 | 9.661e-1 | 1.015e-0 | 1.041e-0 | 1.026e-0 | 9.703e-1 | 8.949e-1 | 9.095e-1 | 1.197e-0 | **2.673e-1** |
| Weather | 5 | 5.629e-1 | 2.394e-1 | 3.965e-1 | 4.307e-1 | 5.859e-2 | 4.510e-1 | 1.770e-1 | 1.063e-1 | **5.682e-2** | 7.222e-2 | 1.055e-1 | 6.979e-2 |
| | **6(PB)** | 5.575e-1 | 2.393e-1 | 3.598e-1 | 4.337e-1 | 5.839e-2 | 4.476e-1 | 1.121e-1 | 9.845e-2 | 5.549e-2 | 9.418e-2 | 1.282e-1 | **4.053e-2** |
| | 7 | 5.533e-1 | 2.395e-1 | 3.623e-1 | 4.397e-1 | 5.953e-2 | 4.471e-1 | 1.240e-1 | 9.803e-2 | 5.571e-2 | 8.327e-2 | 1.482e-1 | **3.451e-2** |
| | 8 | 5.475e-1 | 1.243e-1 | 3.659e-1 | 4.265e-1 | 5.861e-2 | 4.480e-1 | 1.243e-1 | 9.208e-2 | 5.896e-2 | 6.546e-2 | 1.653e-1 | **4.346e-2** |
| Electricity | 7 | 1.655e-2 | 2.437e-2 | 1.183e-2 | 1.676e-2 | 1.213e-2 | **1.002e-2** | 1.498e-2 | 1.651e-2 | 1.732e-2 | 6.070e-2 | 3.251e-2 | 1.440e-2 |
| | **8(PB)** | 1.727e-2 | 2.442e-2 | 1.065e-2 | 1.545e-2 | 1.168e-2 | 9.554e-3 | 1.461e-2 | 1.573e-2 | 1.710e-2 | 6.759e-2 | 3.240e-2 | **5.990e-3** |
| | 9 | 1.617e-2 | 2.449e-2 | 1.043e-2 | 1.505e-2 | 1.058e-2 | 9.548e-3 | 1.325e-2 | 1.512e-2 | 1.292e-2 | 6.279e-2 | 3.066e-2 | **8.540e-3** |
| | 10 | 1.418e-2 | 2.440e-2 | 1.179e-2 | 1.280e-2 | 1.032e-2 | 9.206e-3 | 1.442e-2 | 1.512e-2 | 1.301e-2 | 7.648e-2 | 2.791e-2 | **4.420e-4** |
| Exchange_Rate | 4 | 1.806e-0 | 1.139e-0 | 4.152e-1 | 4.477e-1 | 5.125e-2 | 1.648e-0 | 6.052e-2 | 4.265e-1 | 6.188e-2 | **1.689e-2** | 2.800e-2 | 3.206e-2 |
| | **5(PB)** | 1.792e-0 | 1.136e-0 | 6.595e-1 | 4.523e-1 | 5.768e-2 | 1.630e-0 | 6.519e-2 | 3.779e-1 | 5.707e-2 | 1.552e-2 | 2.824e-2 | **1.092e-2** |
| | 6 | 1.798e-0 | 1.134e-0 | 4.298e-1 | 4.483e-1 | 5.621e-2 | 1.620e-0 | 1.188e-1 | 3.059e-1 | 3.678e-2 | 1.748e-2 | 2.624e-2 | **9.429e-3** |
| Traffic | **6(PB)** | 2.841e-2 | 3.052e-2 | 1.179e-2 | 1.981e-2 | 1.933e-2 | 1.252e-2 | 1.909e-2 | 1.595e-2 | 1.887e-2 | 4.988e-2 | 6.154e-2 | **9.981e-3** |
| | 7 | 2.521e-2 | 3.053e-2 | 1.105e-2 | 1.660e-2 | 1.567e-2 | 1.193e-2 | 1.808e-2 | 1.529e-2 | 1.941e-2 | 4.350e-2 | 5.953e-2 | **9.783e-3** |
| | 8 | 2.839e-2 | 3.054e-2 | 1.126e-2 | 1.625e-2 | 1.560e-2 | 1.643e-2 | 1.643e-2 | 1.643e-2 | 1.911e-2 | 3.871e-2 | 6.358e-2 | **5.651e-3** |
| | 9 | 4.055e-2 | 3.054e-2 | 1.134e-2 | 1.751e-2 | 1.422e-2 | 1.083e-2 | 1.428e-2 | 1.623e-2 | 1.614e-2 | 4.304e-2 | 6.966e-2 | **3.308e-3** |

In conclusion, when OPPCL is guided by a reasonably sized set of frequent OPPs as prior knowledge, the pattern constrained loss can substantially improve the model's prediction accuracy on samples conforming to the pattern paradigm.

## 4.4 ANALYSIS OF PREDICTION RESULTS AT THE "PERTURBATION BOUNDARY"

### 4.4.1 ANALYSIS OF PREDICTION RESULTS FOR RANDOMLY SAMPLED DISCRETE POINTS

During the experiments, we applied the pattern matching mechanism described in Section 3.3 to segment the test set and used random sampling to ensure fairness. We calculated the MSE between the predicted and ground truth values at the end of the windows in the sampled sets of frequent OPP samples, as well as in the sets of non-OPP (three samples were randomly selected for each set). Regarding the choice of pattern length (i.e., window size), we selected lengths that fall on the 'perturbation boundary' as indicated in Table 1 with length nine for the SWaT dataset and length seven for the ETTh1 dataset. The prediction results for the two datasets are presented in Figures 3. As shown in Figure 3, for the SWaT dataset, the predictions made by the OPPCL model trained with pattern constrained loss are noticeably closer to the ground truth values in all three randomly sampled frequent OPP samples. For instance, in the second sample, the difference between prediction and ground truth is only 0.002, whereas the difference w/o CL without the pattern constrained loss is 0.014. In contrast, for the samples from the non-OPP set, the performance of both OPPCL and w/o CL is more susceptible to the randomness of the sampled instances due to the lack of prior pattern knowledge. The analysis of prediction results on randomly sampled discrete points from the ETTh1 dataset is presented in Appendix C.6.

Table 2: MAE Comparison Across Models on Multiple Datasets.

| Dataset | | Autoformer | ConvLSTM | Crossformer | Informer | LSTNet | MegaCRN | SCINet | Leddam | TimeCSL | CycleNet | w/o CL | OPPCL |
|---|---|---|---|---|---|---|---|---|---|---|---|---|---|
| SWaT | 7 | 1.717e-2 | 1.648e-2 | 3.977e-2 | 2.337e-2 | **1.506e-2** | 1.644e-2 | 1.026e-1 | 1.965e-2 | 1.717e-2 | 1.592e-2 | 2.834e-2 | 2.745e-2 |
| | 8 | 1.947e-2 | 1.807e-2 | 1.688e-2 | 2.918e-2 | **1.542e-2** | 1.621e-2 | 1.239e-1 | 1.992e-2 | 1.698e-2 | 1.610e-2 | 2.967e-2 | 2.949e-2 |
| | **9(PB)** | 1947e-2 | 1.767e-2 | 1.264e-1 | 8.657e-2 | 1.527e-2 | 1.611e-2 | 1.652e-1 | 2.058e-2 | 1.609e-2 | 1.608e-2 | 3.078e-2 | **1.487e-2** |
| | 10 | 1686e-2 | 1.828e-2 | 1.639e-2 | 3.067e-2 | 1.494e-2 | 1.589e-2 | 2.512e-1 | 1.979e-2 | 1.696e-2 | 1.570e-2 | 3.269e-2 | **8.989e-3** |
| | 11 | 1.643e-2 | 1.503e-2 | 4.630e-2 | 3.141e-2 | 1.539e-2 | 1.563e-2 | 1.328e-1 | 1.788e-2 | 1.676e-2 | 1.609e-2 | 3.295e-2 | **2.300e-5** |
| | 12 | 2.182e-2 | 1.962e-2 | 3.096e-2 | 9.093e-2 | 1.548e-2 | 1.769e-2 | 2.260e-1 | 1.858e-2 | 1.580e-2 | 1.625e-2 | 3.477e-2 | **6.400e-5** |
| | 13 | 2.058e-2 | 1.648e-2 | 1.201e-1 | 6.826e-2 | 1.491e-2 | 1.515e-2 | 2.528e-1 | 1.837e-2 | 1.962e-2 | 1.650e-2 | 3.451e-2 | **1.647e-3** |
| ETTh1 | 6 | 4.747e-1 | 4.715e-1 | 4.687e-1 | **4.683e-1** | 4.695e-1 | 4.743e-1 | 4.702e-1 | 4699e-1 | 4.705e-1 | 5.122e-1 | 4.882e-1 | 6.433e-1 |
| | **7(PB)** | 4.789e-1 | 4.717e-1 | 4.691e-1 | 4.712e-1 | 4.979e-1 | 4.746e-1 | 4.685e-1 | 4.712e-1 | 4.750e-1 | 5.272e-1 | 4.906e-1 | **4.886e-1** |
| | 8 | 4.814e-1 | 4.721e-1 | 4.711e-1 | 4.732e-1 | 4.688e-1 | 4.749e-1 | 4.683e-1 | 4.727e-1 | 4.771e-1 | 5.241e-1 | 4.960e-1 | **4.167e-1** |
| | 9 | 4.842e-1 | 4.724e-1 | 4.717e-1 | 4.756e-1 | 4.698e-1 | 4.752e-1 | 4.703e-1 | 4.739e-1 | 4.755e-1 | 5.599e-1 | 5.015e-1 | **9.314e-2** |
| ETTh2 | 6 | 3.275e-1 | 3.392e-1 | **3.074e-1** | 3.125e-1 | 3.199e-1 | 3.277e-1 | 3.551e-1 | 3.079e-1 | 3.211e-1 | 3.514e-1 | 3.627e-1 | 3.589e-1 |
| | **7(PB)** | 3.278e-1 | 3.404e-1 | 3.071e-1 | 3.109e-1 | 3.184e-1 | 3.278e-1 | 3.294e-1 | 3.078e-1 | 3.176e-1 | 3.547e-1 | 3.813e-1 | **2.539e-1** |
| | 8 | 3.285e-1 | 3.416e-1 | 3.094e-1 | 3.082e-1 | 3.182e-1 | 3.279e-1 | 3.474e-1 | 3.097e-1 | 3.166e-1 | 3.559e-1 | 3.888e-1 | **1.053e-1** |
| ETTm1 | 6 | 2.627e-1 | 2.570e-1 | 2.575e-1 | 2.571e-1 | 2.552e-1 | 2.578e-1 | 2.808e-1 | **2.548e-1** | 2.556e-1 | 2.549e-1 | 2.744e-1 | 3.337e-1 |
| | **7(PB)** | 2.638e-1 | 2.570e-1 | 2.591e-1 | 2.579e-1 | 2.551e-1 | 2579e-1 | 2.800e-1 | 2.551e-1 | 2.551e-1 | 2.557e-1 | 2.751e-1 | **1.560e-1** |
| | 8 | 2.638e-1 | 2.570e-1 | 2.594e-1 | 2.563e-1 | 2.553e-1 | 2.579e-1 | 2.654e-1 | 2.558e-1 | 2.566e-1 | 2.562e-1 | 2.787e-1 | **4.540e-2** |
| | 9 | 2.645e-1 | 2.570e-1 | 2.600e-1 | 2.572e-1 | 2.555e-1 | 2.579e-1 | 2.726e-1 | 2.558e-1 | 2.563e-1 | 2.567e-1 | 2.783e-1 | **2.023e-3** |
| ETTm2 | 6 | 2.074e-1 | 2.011e-1 | 1.983e-1 | **1.966e-1** | 2.034e-1 | 2.291e-1 | 3.438e-1 | 2.034e-1 | 2.196e-1 | 2.005e-1 | 2.523e-1 | 3.651e-1 |
| | 7 | 2.053e-1 | 2.011e-1 | **1.982e-1** | 2.095e-1 | 2.018e-1 | 2.264e-1 | 2.924e-1 | 2.073e-1 | 2.132e-1 | 2.001e-1 | 2.600e-1 | 2.698e-1 |
| | **8(PB)** | 2.032e-1 | 2.011e-1 | 1.969e-1 | 2.040e-1 | 2.021e-1 | 2.211e-1 | 3.728e-1 | 2.269e-1 | 2.108e-1 | 2.012e-1 | 2.698e-1 | **8.584e-2** |
| | 9 | 2.029e-1 | 2.010e-1 | 2.060e-1 | 2.043e-1 | 2.025e-1 | 2.199e-1 | 2.928e-1 | 2.077e-1 | 2.049e-1 | 2.037e-1 | 2.702e-1 | **7.630e-4** |
| PowerSystem | 4 | **2.811e-1** | 2.928e-1 | 2.840e-1 | 2.898e-1 | 2.906e-1 | 2.933e-1 | 3.041e-1 | 2.831e-1 | 2.818e-1 | 2.942e-1 | 3.464e-1 | 3.249e-1 |
| | 5 | **2.794e-1** | 2.930e-1 | 2.868e-1 | 2.949e-1 | 2.914e-1 | 2.932e-1 | 3.065e-1 | 2.868e-1 | 2.804e-1 | 2.931e-1 | 3.601e-1 | 2.930e-1 |
| | **6 (PB)** | 2.799e-1 | 2.925e-1 | 2.882e-1 | 2.967e-1 | 2.940e-1 | 2.923e-1 | 3.004e-1 | 2.875e-1 | 2.778e-1 | 2.926e-1 | 3.680e-1 | **2.303e-1** |
| | 7 | 2.774e-1 | 2.918e-1 | 2.793e-1 | 2.890e-1 | 2.913e-1 | 2.919e-1 | 3.011e-1 | 2.930e-1 | 2.763e-1 | 2.929e-1 | 3.756e-1 | **2.129e-1** |
| Weather | 5 | 3.208e-1 | 2.557e-1 | 2.680e-1 | 2.942e-1 | **1.309e-1** | 2.824e-1 | 3.098e-1 | 1.777e-1 | 1.364e-1 | 1.708e-1 | 2.192e-1 | 1.759e-1 |
| | **6(PB)** | 3.184e-1 | 2.556e-1 | 2.477e-1 | 2.852e-1 | 1.304e-1 | 2.817e-1 | 2.449e-1 | 1.722e-1 | 1.292e-1 | 2.034e-1 | 2.439e-1 | **1.217e-1** |
| | 7 | 3.169e-1 | 2.558e-1 | 2.451e-1 | 2.836e-1 | 1.321e-1 | 2.813e-1 | 2.461e-1 | 1.758e-1 | 1.322e-1 | 1.923e-1 | 2.697e-1 | **1.298e-1** |
| | 8 | 3.162e-1 | 2.606e-1 | 2.515e-1 | 2.759e-1 | 1.298e-1 | 2.821e-1 | 2.606e-1 | 1.708e-1 | 1.342e-1 | 1.509e-1 | 2.913e-1 | **1.243e-1** |
| Electricity | 7 | 8.959e-2 | 1.151e-1 | 6.986e-2 | 8.558e-2 | 6.853e-2 | **6.027e-2** | 9.243e-2 | 8.690e-2 | 8.572e-2 | 1.961e-1 | 1.372e-1 | 9.095e-2 |
| | **8(PB)** | 9.068e-2 | 1.152e-2 | 6.514e-2 | 8.193e-2 | 6.748e-2 | 5.767e-2 | 9.102e-2 | 8.523e-2 | 8.763e-2 | 2.130e-1 | 1.274e-1 | **5.082e-2** |
| | 9 | 8.612e-2 | 1.154e-1 | 6.522e-2 | 8.186e-2 | 6.393e-2 | 5.774e-2 | 8.418e-2 | 8.484e-2 | 7.115e-2 | 1.992e-1 | 1.158e-1 | **5.603e-2** |
| | 10 | 7.816e-2 | 1.151e-1 | 7.400e-2 | 7.672e-2 | 6.345e-2 | 5.767e-2 | 8920e-2 | 8679e-2 | 7.580e-2 | 2.251e-1 | 1091e-1 | **1.345e-2** |
| Exchange_Rate | 4 | 9.803e-1 | 8.470e-1 | 4.330e-1 | 4.550e-1 | 1.726e-1 | 9.553e-1 | 2.244e-1 | 4.432e-1 | 2.048e-1 | **1.073e-1** | 1.284e-1 | 1.409e-1 |
| | **5(PB)** | 9.755e-1 | 8.460e-1 | 6.242e-1 | 4.623e-1 | 1.809e-1 | 9.484e-1 | 2.306e-1 | 4.188e-1 | 1.891e-1 | 9.924e-2 | 1.298e-1 | **7.984e-2** |
| | 6 | 9.781e-1 | 8.451e-1 | 4.418e-1 | 4.552e-1 | 1.779e-1 | 9.447e-1 | 3.135e-1 | 3.755e-1 | 1.518e-1 | 1.096e-1 | 1.260e-1 | **7.290e-2** |
| Traffic | **6(PB)** | 1.173e-1 | 1.261e-1 | 7.617e-2 | 1.021e-1 | 9.569e-2 | 7.814e-2 | 1.036e-1 | 9.340e-2 | 9.703e-2 | 1.735e-1 | 1.776e-1 | **7.092e-2** |
| | 7 | 1.130e-1 | 1.262e-1 | 7.398e-2 | 9.134e-2 | 8.871e-2 | 7.578e-2 | 9.931e-2 | 9.141e-2 | 9.775e-2 | 1.614e-1 | 1.761e-1 | **6.729e-2** |
| | 8 | 1.178e-1 | 1.262e-1 | 7.358e-2 | 9.238e-2 | 8.880e-2 | 9.483e-2 | 9.483e-2 | 9.379e-2 | 1.001e-1 | 1.550e-1 | 1.816e-1 | **5.066e-2** |
| | 9 | 1.375e-1 | 1.262e-1 | 7.323e-2 | 9.535e-2 | 8.546e-2 | 7.270e-2 | 8.402e-2 | 9.328e-2 | 9.113e-2 | 1.624e-1 | 1.847e-1 | **3.947e-2** |

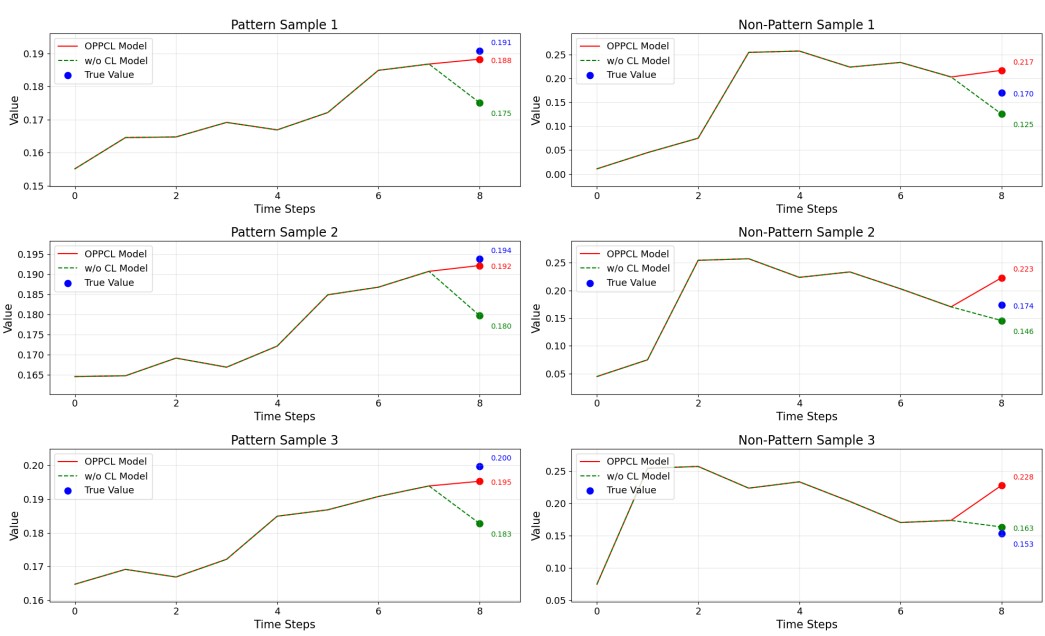

Figure 3: Prediction comparison on three randomly sampled instances from two sets on the SWaT dataset

To further analyze the differences in overall trend prediction, prediction latency, and turning point capture between OPPCL and w/o CL on consecutively matching pattern samples, we extend our

evaluation beyond the previous random sampling results.The detailed prediction results and analysis for consecutive matching patterns are presented in Appendix C.7.

### 4.4.2 COMPARATIVE ANALYSIS OF ATTENTION ENTROPY

To further investigate the impact of the pattern-constrained loss on the model's attention mechanism, we extracted the attention distributions of the trained OPPCL and w/o CL models on the SWaT dataset. Specifically, we computed the attention weights from the prediction position to all input positions and calculated the attention entropy for each attention head, as well as the average attention entropy $H_{avg}$(A detailed description of this metric is presented in Appendix C.2.) across all heads. The distribution and average value comparisons of attention entropy for both models are shown in Figure 4. Figures 4(a) and 4(b) show that the average attention entropy for frequent OPP samples in

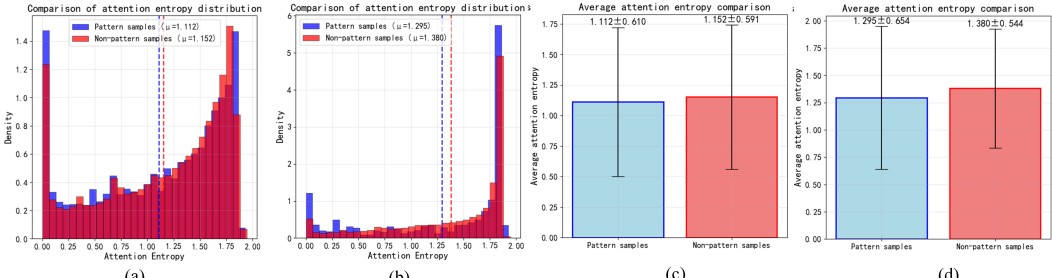

Figure 4: Comparison of attention entropy between OPPCL and w/o CL after training on the SWaT dataset, focusing on frequent OPP samples and non-OPP samples. Subfigures (a) and (c) correspond to OPPCL, while (b) and (d) correspond to w/o CL.

OPPCL is $\mu = 1.112$, which is notably lower than that of w/o CL ($\mu = 1.295$). Furthermore, the bar charts in Figures 4(c)(d) also demonstrate lower $y$-axis values for frequent OPP samples under OPPCL, indicating more focused attention. The comparative analysis of attention entropy for the ETTh1 dataset is presented in Appendix C.8. These results suggest that training with the pattern-constrained loss enables the model to learn a more concentrated attention distribution over frequent OPP samples, indicating a stronger alignment between the model's attention and the encoded prior knowledge.

Finally, to assess whether this enhanced attention concentration is consistent across the distribution, rather than merely reflected in a lower mean entropy, we further compared the quantile-based distributions of attention entropy for both models. The detailed results and analysis are presented in Appendix C.9.

## 5 CONCLUSIONS

This paper proposes OPPCL, a novel time series forecasting model that incorporates frequent OPPs as structural priors to enhance model performance in pattern-rich regions. Unlike traditional forecasting methods that rely solely on numerical regression, OPPCL is the first to explicitly model relative ordering patterns within time series and introduces a position-sensitive pattern-constrained loss function. This loss guides the model to generate predictions that align with the ordering logic of frequent patterns. Based on this insight, we develop a complete training framework that integrates dimensionality reduction, pattern mining, pattern matching, and structure-aware optimization. Experiments conducted on ten datasets demonstrate that OPPCL not only outperforms mainstream baseline models in terms of overall prediction error, but also achieves significantly better performance on structurally patterned samples. Furthermore, OPPCL enhances the attention focus of baseline Transformer-based models, indicating improved interpretability.

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

# A  PRELIMINARY

**Definition 1.** Let a time series be denoted by $\mathbf{t} = (t_1, t_2, \ldots, t_n)$, where $t_i$ represents the observed value at the $i$-th time point.

**Definition 2.** For a pattern $\mathbf{p} = (p_1, p_2, \ldots, p_m)$ with length $m$, its OPP is a relative order vector $R(\mathbf{p}) = \big(\mathrm{rank}_{\mathbf{p}}(p_1),\ \mathrm{rank}_{\mathbf{p}}(p_2),\ \ldots,\ \mathrm{rank}_{\mathbf{p}}(p_m)\big)$, where $\mathrm{rank}_{\mathbf{p}}(p_i)$ is the ascending rank of $p_i$ within pattern $\mathbf{p}$.

**Example 1.** Given $\mathbf{p} = (31, 27, 33, 30)$, the relative order vector is $R(\mathbf{p}) = (3, 1, 4, 2)$, since $(31, 27, 33, 30)$ sorts to $27 < 30 < 31 < 33$, i.e., 31 is the third smallest. Similarly, 27, 33, and 30 are the first, fourth, and second smallest, respectively.

**Definition 3.** The support of pattern $\mathbf{p}$ in time series $\mathbf{t}$, denoted by $sup(\mathbf{p}, \mathbf{t})$, is defined as the number of non-overlapping sub-time series in $\mathbf{t}$ that match the OPP $\mathbf{p}$.

**Example 2.** Given a time series $\mathbf{t} = (24, 31, 27, 33, 30, 24, 21, 25, 23, 26, 22, 27, 24, 28, 23, 29)$ and pattern $\mathbf{p} = (3, 1, 4, 2)$, the matching process is as follows:

1. Sub-time series 1: $(t_2, t_3, t_4, t_5) = (31, 27, 33, 30)$ has $R(\cdot) = (3, 1, 4, 2)$; match successful.

2. Sub-time series 2: $(t_6, t_7, t_8, t_9) = (24, 21, 25, 23)$ has $R(\cdot) = (3, 1, 4, 2)$; match successful.

3. Sub-time series 3: $(t_{10}, t_{11}, t_{12}, t_{13}) = (26, 22, 27, 24)$ has $R(\cdot) = (3, 1, 4, 2)$; match successful.

Thus, $sup(\mathbf{p}, \mathbf{t}) = 3$.

**Definition 4.** A pattern $\mathbf{p}$ is a frequent OPP if $sup(\mathbf{p}, \mathbf{t}) \geq minsup$, where $minsup$ is a user-specified minimum support threshold.

**Example 3.** Given a threshold $minsup = 3$, from Example 2, pattern $\mathbf{p} = (3, 1, 4, 2)$ is a frequent OPP, since $sup(\mathbf{p}, \mathbf{t}) = minsup = 3$, which meets the minimum requirement.

**Definition 5.** The prefix OPP of pattern $\mathbf{p}$ is $prefix(\mathbf{p}) = R(p_1, p_2, \ldots, p_{m-1})$, i.e., the relative order of the pattern excluding its last element. Similarly, the suffix OPP is $suffix(\mathbf{p}) = R(p_2, p_3, \ldots, p_m)$, i.e., the relative order excluding the first element.

**Example 4.** For $\mathbf{p} = (3, 1, 4, 2)$, we have $prefix(\mathbf{p}) = R(3, 1, 4) = (2, 1, 3)$, $suffix(\mathbf{p}) = R(1, 4, 2) = (1, 3, 2)$.

**Definition 6.** Given two patterns $\mathbf{p}$ and $\mathbf{q}$, each of length $m$, if $R(suffix(\mathbf{p})) = R(prefix(\mathbf{q}))$, then $\mathbf{p}$ and $\mathbf{q}$ can be combined to generate a super-pattern of length $m + 1$. Pattern Generation Rules (Pattern Fusion):

1. **Case 1:** If $p_1 \neq q_m$, then a single super-pattern $\mathbf{r} = \mathbf{p} \oplus \mathbf{q}$ is generated by concatenating $\mathbf{p}$ and $\mathbf{q}$, followed by re-ranking the combined sequence to obtain the final OPP.

2. **Case 2:** If $p_1 = q_m$, then two super-patterns $\mathbf{r}$ and $\mathbf{h}$ are generated: $\mathbf{r} = \mathbf{p} \oplus \mathbf{q}$, $\mathbf{h} = \mathbf{q} \oplus \mathbf{p}$. Both OPPs are then re-ranked accordingly.

**Example 5.** Given $\mathbf{p} = (3, 1, 2)$ and $\mathbf{q} = (1, 3, 2)$, since $p_1 \neq q_m$, we generate: $\mathbf{r} = (4, 1, 3, 2)$.

Given $\mathbf{p} = (2, 1, 3)$ and $\mathbf{q} = (1, 3, 2)$, since $p_1 = q_m$, we generate: $\mathbf{r} = \mathbf{p} \oplus \mathbf{q} = (3, 1, 4, 2)$, $\mathbf{h} = \mathbf{q} \oplus \mathbf{p} = (2, 1, 4, 3)$.

**Definition 7.** Given a window size $v$, a sliding window refers to a sub-time series of length $v$ that moves across the time series to extract sub-time series. Specifically, if the sliding window starts at position $i$, the extracted sub-time series is: $window(i) = (t_i, t_{i+1}, \ldots, t_{i+v-1})$, i.e., each sliding window captures $v$ consecutive time points from the original series.

**Definition 8.** Given the first $k$ observations of a time series $\mathbf{t} = (t_1, t_2, \ldots, t_k)$, the goal is to predict the value at the next time step $t_{k+1}$, which is formally defined as follows: $\hat{t}_{k+1} = f(t_1, t_2, \ldots, t_k)$, where $\hat{t}_{k+1}$ denotes the predicted value, and $f$ is a mapping function learned by the model.

**Definition 9.** Let the input time series, after sliding window processing, have an effective length of $v - 1$. The input at each time step $i$ is represented by a $d$-dimensional vector $\mathbf{z}_i$, resulting in

the entire input being represented as a matrix: $Z = (\mathbf{z}_1, \mathbf{z}_2, \ldots, \mathbf{z}_{v-1})$, $Z \in \mathbb{R}^{(v-1) \times d}$, where $d$ is the embedding dimension. The input matrix $Z$ is projected into the query, key, and value spaces using three sets of learnable parameters $W_Q, W_K, W_V \in \mathbb{R}^{d \times d}$, defined as: $Q = ZW_Q$, $K = ZW_K$, $V = ZW_V$. The attention weights are computed as: $A = \mathrm{softmax}\left(\frac{QK^\top}{\sqrt{d}}\right)$, where $A$ is the attention matrix, representing the attention score of each time step with respect to others. The updated feature representation is computed as: $Z' = AV$, $Z' \in \mathbb{R}^{(v-1) \times d}$.

**Definition 10.** Our objective is to incorporate OPPs as prior knowledge by designing a pattern-constrained loss function. This enables the model to capture and learn the OPPs embedded in the time series $\mathbf{t}$, thereby ensuring order preservation during prediction. By leveraging these structural constraints, the model generates predictions that are not only accurate in terms of MSE but also consistent with the underlying order patterns, ultimately reducing the prediction error on pattern-conforming samples.

# B ALGORITHM PSEUDOCODE

## B.1 DIMENSIONALITY REDUCTION PROCESS

The pseudocode of the dimensionality reduction process in Section 3.1 is shown in Algorithm 1.

---

**Algorithm 1: CNN training and reduced-dimensional feature extraction**

**Input:** Time series $S \in \mathbb{R}^{n \times d}$; flag $return\_intermediate$ (default False); number of conv modules $n_{\mathrm{conv}}$; number of fully connected layers $n_{\mathrm{fc}}$; number of classes $c$.
**Output:** Final output $S' \in \mathbb{R}^{n \times c}$ or intermediate output. $\mathbf{S}_{\mathrm{inter}} \in \mathbb{R}^{n \times 1}$
1: $H \leftarrow S$;
2: $H \leftarrow \mathrm{Reshape}(H, [n, 1, d])$;
3: **for** $i = 1$ **to** $n_{\mathrm{conv}}$ **do**
4:    Assign parameters $F\_i$ and $k\_i$ to the $i$-th convolutional module;
5:    $H \leftarrow \mathrm{Conv1D}; (H, \mathrm{filters} = F\_i, \mathrm{kernel\_size} = k\_i)$;
6:    $H \leftarrow \mathrm{ReLU}(H)$;
7:    Assign parameter $p\_size$ to pooling module;
8:    $H \leftarrow \mathrm{MaxPool1D}(H, p_{\mathrm{size}})$;
9: **end for**
10: $H_{\mathrm{flat}} \leftarrow \mathrm{Reshape}(H, [n, -1])$;
11: $H_{\mathrm{inter}} \leftarrow H_{\mathrm{flat}}$;
12: **for** $j = 1$ **to** $n_{\mathrm{fc}} - 1$ **do**
13:    Assign parameter $d\_j$ to the $j$-th fully connected layer in each iteration(The output dim of $(n_{\mathrm{fc}} - 1)$-th layer is 1);
14:    $H_{\mathrm{inter}} \leftarrow \mathrm{Linear}(H_{\mathrm{inter}}, \mathrm{output\_dim} = d\_j)$;
15:    $H_{\mathrm{inter}} \leftarrow \mathrm{ReLU}(H_{\mathrm{inter}})$;
16: **end for**
17: $\mathbf{S}_{\mathrm{inter}} \leftarrow H_{\mathrm{inter}}$;
18: $S' \leftarrow \mathrm{Linear}(H_{\mathrm{inter}}, \mathrm{output\_dim} = c)$;
19: **if** $return\_intermediate = \mathrm{True}$ **then**
20:    **return** $\mathbf{S}_{\mathrm{inter}}$;
21: **else**
22:    **return** $S'$;
23: **end if**

---

## B.2 EXTRACTING FREQUENT OPPS

The pseudocode of OPP-Mining for prior knowledge extraction in Section 3.2 is shown in Algorithm 2.

---

**Algorithm 2: Opp-Mining for Extracting Frequent OPPs**

---

**Input:** Time series $\mathbf{S}_{\text{inter}} = (s_{\text{inter}_1}, s_{\text{inter}_2}, \ldots, s_{\text{inter}_n}) \in \mathbb{R}^{n \times 1}$; minimum support threshold $minsup$; expected sliding window size $w$; window dynamic size $l$; step size $step$; sliding window sub-time series set $W_l$; an OPP with length $l$, $\mathbf{p} \in S_l$; support counter $supp(\mathbf{p})$.

**Output:** Frequent OPPs with length $v$, $F_v$, and corresponding support values $support\_value$.

1: $l \leftarrow 2$, $W_l \leftarrow \emptyset$;
2: **while** $l \leq v$ **do**
3:    **for** $i = 1$ **to** $n - l + 1$ **with increment** $step$ **do**;
4:       Extract sub-time series $\mathbf{S}^{(i)} \leftarrow (s_{\text{inter}_i}, s_{\text{inter}_{i+1}}, \ldots, s_{\text{inter}_{i+l-1}})$;
5:       Add $\mathbf{S}^{(i)}$ to $W_l$;
6:    **end for**
7:    **for each** $\mathbf{S}^{(i)} \in W_l$ **do**
8:       Compute OPP $R(\mathbf{S}^{(i)})$;
9:       Increment $supp(R(\mathbf{S}^{(i)}))$ by 1;
10:    **end for**
11:    Construct frequent OPP set: $F_l \leftarrow \{ \mathbf{p} \in S_l \mid supp(\mathbf{p}) \geq minsup \}$;
12:    **if** $l < v$ **then**;
13:       Initialize $F_{l+1} \leftarrow \emptyset$;
14:       **for each** pair $(\mathbf{p}, \mathbf{q}) \in F_l \times F_l$ **do**
15:         **if** $R(suffix(\mathbf{p})) = R(prefix(\mathbf{q}))$ **then**
16:           Generate candidate super-pattern $\mathbf{r}$ (or $(\mathbf{r}, \mathbf{h})$);
17:           Calculate $supp(\mathbf{r})$;
18:           **if** $supp(\mathbf{r}) > minsup$ **then**
19:              Add $\mathbf{r}$ to $F_{l+1}$;
20:           **end if**
21:         **end if**
22:       **end for**
23:    **end if**
24:    $l \leftarrow l + 1$;
25: **end while**
26: **if** $l = v$ **then**
27:    Calculate $support\_value \leftarrow [ supp(\mathbf{p}) \mid \mathbf{p} \in F_v ]$;
28: **end if**
29: **return** $F_v$, $support\_value$;

---

### B.3 THE LOSS COMPUTATION PROCESS

The pseudocode of the loss computation process of OPPCL in Section 3.4 is shown in Algorithm 3.

---

**Algorithm 3: Calculation method of loss function**

---

**Input:** Non-OPP sample set $D_{\text{non}-\text{pattern}}$, frequent OPPs sample set $D_{\text{pattern}}$, set of frequent OPPs with position markers $F_v = \{(\mathbf{p}_i, sv_i)\}_{i=1}^{|F_v|}$, predicted value $\hat{y}$, true value $y$, sequence fragment $\mathbf{X}$, sequence formed by concatenating $\mathbf{X}$ and $\hat{y}$ denoted by $\hat{\mathbf{X}}$, hyperparameters $\epsilon, \lambda_1, \lambda_2, \lambda_3$.

**Output:** Base prediction loss $\mathcal{L}_{\text{basic}}$, pattern area prediction loss $\mathcal{L}_{\text{pattern}}$, pattern constraint loss $\mathcal{L}_{\text{constraint}}$, total loss $\mathcal{L}_{\text{total}}$.

1: $\mathcal{L}_{\text{total}} \leftarrow 0$, $\mathcal{L}_{\text{basic}} \leftarrow 0$, $\mathcal{L}_{\text{pattern}} \leftarrow 0$, $\mathcal{L}_{\text{constraint}} \leftarrow 0$;
2: $\mathcal{L}_{\text{basic}} \leftarrow \frac{1}{|D_{\text{non}-\text{pattern}}|} \sum_{(\mathbf{X},\hat{y}) \in D_{\text{non}-\text{pattern}}} (\hat{y} - y)^2$;
3: $\mathcal{L}_{\text{pattern}} \leftarrow \frac{1}{|D_{\text{pattern}}|} \sum_{(\mathbf{X},\hat{y}) \in D_{\text{pattern}}} (\hat{y} - y)^2$;
4: $st \leftarrow \sum_{i=1}^{|F_v|} sv_i$;
5: **for** each pattern $i = 1$ **to** $|F_v|$ **do**

---

6:    **if** $\mathbf{p}_i[-1] = 0$ **then**
7:      Set right reference $ref_r \leftarrow \hat{\mathbf{X}}[1]$;
8:      $loss_i \leftarrow \frac{1}{|D_{\text{pattern}}|} \sum_{(\mathbf{X}, \hat{y}) \in D_{\text{pattern}}} \frac{sv_i}{st} \max(0, \hat{y} - ref_r + \epsilon)$;
9:      $\mathcal{L}_{\text{constraint}} \leftarrow \mathcal{L}_{\text{constraint}} + loss_i$;
10:   **else if** $\mathbf{p}_i[-1] = v - 1$ **then**
11:     Set left reference $ref_l \leftarrow \hat{\mathbf{X}}[v-2]$;
12:     $loss_i \leftarrow \frac{1}{|D_{\text{pattern}}|} \sum_{(\mathbf{X}, \hat{y}) \in D_{\text{pattern}}} \frac{sv_i}{st} \max(0, ref_l - \hat{y} + \epsilon)$;
13:     $\mathcal{L}_{\text{constraint}} \leftarrow \mathcal{L}_{\text{constraint}} + loss_i$;
14:   **else if** $0 < \mathbf{p}_i[-1] < v - 1$ **then**
15:     Set left and right references: $ref_l \leftarrow \hat{\mathbf{X}}[\mathbf{p}_i[-1] - 1], ref_r \leftarrow \hat{\mathbf{X}}[\mathbf{p}_i[-1] + 1]$;
16:     $loss_i \leftarrow \frac{1}{|D_{\text{pattern}}|} \sum_{(\mathbf{X}, \hat{y}) \in D_{\text{pattern}}} \big( \max(0, ref_l - \hat{y} + \epsilon) + \max(0, \hat{y} - ref_r + \epsilon) \big)$;
17:     $\mathcal{L}_{\text{constraint}} \leftarrow \mathcal{L}_{\text{constraint}} + loss_i$;
18:   **end if**
19: **end for**
20: Compute total loss: $\mathcal{L}_{\text{total}} \leftarrow \lambda_1 \mathcal{L}_{\text{basic}} + \lambda_2 \mathcal{L}_{\text{pattern}} + \lambda_3 \mathcal{L}_{\text{constraint}}$;
21: **return** $\mathcal{L}_{\text{basic}}, \mathcal{L}_{\text{pattern}}, \mathcal{L}_{\text{constraint}}, \mathcal{L}_{\text{total}}$;

To better explain the above procedure, we provide an example of a constraint violation below.

**Example 6.** Assume the time series length is $v = 5$. The predicted time series is given by:

$$\hat{\mathbf{X}} = [s_{\text{inter}_0}, s_{\text{inter}_1}, s_{\text{inter}_2}, s_{\text{inter}_3}, \hat{y}].$$

Set $\epsilon = 0.01$. Suppose $F_v^i = (1, 2, 3, 4, 0)$ is a pattern in the pattern set $F_v$, where $F_v^i[-1] = 0$, satisfying Case 1, i.e., the right-boundary constraint. In this case, the model is required to satisfy the constraint:

$$\hat{y} < s_{\text{inter}_0} - \epsilon.$$

Given a sample:

$$\hat{\mathbf{X}} = [0.8, 0.9, 1.0, 1.1, \hat{y} = 0.85].$$

Here, $s_{\text{inter}_0} = 0.8$, then the condition becomes

$$\hat{y} < 0.79 \quad (\text{since } 0.8 - 0.01 = 0.79).$$

However, the current value of $\hat{y} = 0.85$, and clearly $0.85 > 0.79$, thus violating the right-boundary constraint.

The resulting right-boundary constraint loss is calculated as:

$$\mathcal{L}_{\text{constraint}}^r = \max(0, \hat{y} - s_{\text{inter}_0} + \epsilon) = \max(0, 0.85 - 0.8 + 0.01) = 0.06.$$

Therefore, in this example, the predicted value $\hat{y}$ exceeds the required right-boundary, resulting in a non-zero right-boundary constraint loss. This loss serves as a feedback signal to guide the model to adjust its output during training, thereby encouraging better compliance with the OPP.

## C   DETAILS OF EXPERIMENTS

### C.1   DIMENSIONALITY REDUCTION DETAILS

OPPCL applies CNN-based dimensionality reduction to the SWaT dataset and PowerSystem dataset, For the remaining eight datasets we employs PCA to reduce the data at each timestamp to 1 dimension. In this study, we adopt differentiated dimensionality reduction strategies for different types of datasets. The core consideration is not the tools themselves but whether stable, structured representations can be obtained using supervisory signals. For labeled time series data such as SWaT and PowerSystem, which provide operation states or event labels at each time point, we are able to train a CNN as a supervised feature extractor. Through label supervision, the CNN learns discriminative features related to system state changes, allowing it to preserve not only the main trends but also information related to anomalous states during dimensionality reduction. Therefore, using CNN-based dimensionality reduction is appropriate for these datasets. For unlabeled time series

data (the remaining eight datasets), these datasets lack supervisory information at each time point. Using CNN for unsupervised training in this context would make the model's dimensionality reduction directions uncontrollable, potentially introducing biases and even learning noise unrelated to OPP mining. In this case, PCA, as a classical unsupervised linear dimensionality reduction method, provides stable and unbiased principal component directions for unlabeled data, ensuring that OPP mining operates in a consistent and interpretable low-dimensional space.

Table 3: Statistics of all datasets used in our experiments.

| Dataset | Timesteps | Features | Frequency | Training datasets | Test datasets | Label |
|---|---|---|---|---|---|---|
| SWaT | 449,919 | 51 | 1 sec | 359,935 | 89,984 | Yes |
| ETTh1 | 17,420 | 7 | 1 hour | 10,452 | 6,968 | No |
| ETTh2 | 17,420 | 7 | 1 hour | 10,452 | 6,968 | No |
| ETTm1 | 69,680 | 7 | 15 min | 41,808 | 27,873 | No |
| ETTm2 | 69,680 | 7 | 15 min | 41,808 | 27,873 | No |
| PowerSystem | 14,898 | 128 | - | 8,938 | 5,960 | Yes |
| Weather | 26,200 | 21 | 10 min | 15,720 | 10,480 | No |
| Electricity | 26,304 | 320 | 1 hour | 15,782 | 10,522 | No |
| Exchange_Rate | 7,588 | 8 | 1 day | 4,552 | 3,036 | No |
| Traffic | 17,544 | 862 | 1 hour | 10,526 | 7,018 | No |

## C.2 Evaluation Metrics

1) **Mean Squared Error (MSE).** MSE is a widely used evaluation metric in regression analysis and statistical learning. It quantifies the average squared difference between the predicted values and their corresponding ground truth values. Formally, given a set of $n$ predicted values $\hat{y}_i$ and their corresponding true values $y_i$, the MSE is defined as:

$$\text{MSE} = \frac{1}{n}\sum_{i=1}^{n}\left(\hat{y}_i - y_i\right)^2 \tag{16}$$

Due to the squaring operation, this metric penalizes larger deviations more severely than smaller ones, making it particularly sensitive to outliers. The squaring also ensures the metric is always non-negative—a lower value indicates higher prediction accuracy.

2) **Mean Absolute Error (MAE).** MAE is another fundamental evaluation metric in regression analysis and predictive modeling. It measures the average magnitude of the errors between predicted values and ground truth values, without considering their direction. Specifically, for a set of $n$ predictions $\hat{y}_i$ and their corresponding true values $y_i$, the MAE is defined as:

$$\text{MAE} = \frac{1}{n}\sum_{i=1}^{n}\left|\hat{y}_i - y_i\right| \tag{17}$$

Unlike MSE, MAE uses the absolute difference rather than squaring the error term, which results in a metric that is linearly sensitive to the magnitude of deviations. This makes MAE more robust to outliers, as large errors are not amplified quadratically. MAE is always non-negative, and smaller values correspond to better predictive performance.

3) **Entropy of Attention Weights.** This metric measures the 'dispersion' or 'certainty' of the attention distribution. A higher entropy value indicates more dispersed attention, while a lower value suggests that attention is concentrated on fewer positions.

Let $A \in \mathbb{R}^{n \times n}$ denote the attention matrix output from a single attention head. To quantify the uniformity of the attention distribution, we define the entropy of the attention head as:

$$H(A) = -\sum_{i=1}^{n}\sum_{j=1}^{n} A_{ij} \log\left(A_{ij} + \gamma\right) \tag{18}$$

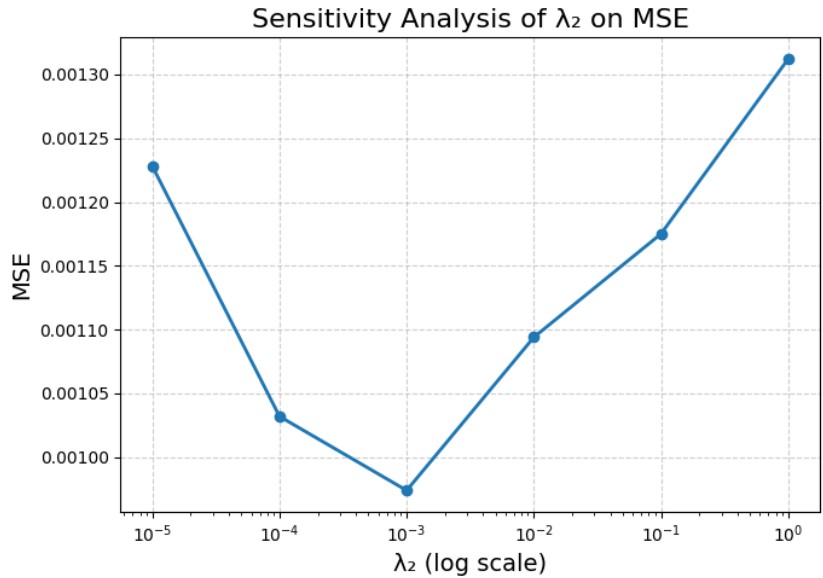

Figure 5: Sensitivity Analysis of $\lambda_2$ on SWaT dataset

where $A_{ij}$ represents the attention weight from the $i$-th query to the $j$-th key, and $\gamma$ is a small constant (set to $1 \times 10^{-10}$ in experiments) to ensure numerical stability.

Finally, the average entropy across multiple attention heads is computed as:

$$H_{\text{avg}} = \frac{1}{n} \sum_{i=1}^{n} H\left(A^{(i)}\right).$$

(19)

### C.3 OPPCL Model and Training Configuration

The embedding dimension of OPPCL is set to 64. The multi-head self-attention mechanism employs four attention heads. The feed-forward network consists of three fully connected layers with output dimensions of 100, 50, and 64. The final fully connected layer, which performs dimensionality compression, outputs a vector of size one to match the input dimension. Additionally, the training parameters used for OPPCL are summarized in Table 4. In OPPCL, the choice of $\lambda_2$ is not arbitrary but follows the following principles:

- The proportion of pattern regions is typically smaller than that of non-pattern regions. If $\lambda_2$ is too small, the MSE in pattern regions will be completely overshadowed by the main MSE, making it almost impossible for the model to perceive gradient signals from pattern regions during the early training stage.

- Conversely, if $\lambda_2$ is set too large, the relative-order constraints (boundary constraints) in the constraint loss will lose nearly all of their influence during gradient descent. Therefore, we select a balanced value for this hyperparameter.

We additionally evaluate the sensitivity of $\lambda_2$ using a predefined list of candidate values. The results for the SWaT and ETTh1 datasets are shown in Figures 5 and 6. As illustrated, when $\lambda_2 = 0.001$, the MSE between predictions and ground truth reaches the lowest value among all candidates. For the remaining datasets, the selection of $\lambda_2$ follows the same rationale.

Table 4: The training parameters used for OPPCL.

| Dataset | Tolerance interval $\epsilon$ | $minsup$ | Batch size | Epochs | Learning Rate | $\lambda_1$ | $\lambda_2$ | $\lambda_3$ |
|---|---|---|---|---|---|---|---|---|
| SWaT | 0.00001 | 450 | 256 | 100 | 0.001 | 1.0 | 0.001 | 1.0 |
| ETTh1 | 0.00001 | 50 | 256 | 150 | 0.0001 | 1.0 | 0.001 | 1.0 |
| ETTh2 | 0.00001 | 20 | 256 | 150 | 0.0001 | 1.0 | 0.001 | 1.0 |
| ETTm1 | 0.00001 | 500 | 256 | 150 | 0.0001 | 1.0 | 0.001 | 1.0 |
| ETTm2 | 0.00001 | 100 | 256 | 150 | 0.0001 | 1.0 | 0.001 | 1.0 |
| PowerSystem | 0.00001 | 60 | 256 | 150 | 0.0001 | 1.0 | 0.001 | 1.0 |
| Weather | 0.00001 | 200 | 256 | 150 | 0.0001 | 1.0 | 0.001 | 1.0 |
| Electricity | 0.00001 | 1200 | 256 | 150 | 0.0001 | 1.0 | 0.01 | 1.0 |
| Exchange_Rate | 0.00001 | 200 | 256 | 150 | 0.0001 | 1.0 | 0.1 | 1.0 |
| Traffic | 0.00001 | 400 | 256 | 150 | 0.0001 | 1.0 | 0.1 | 1.0 |

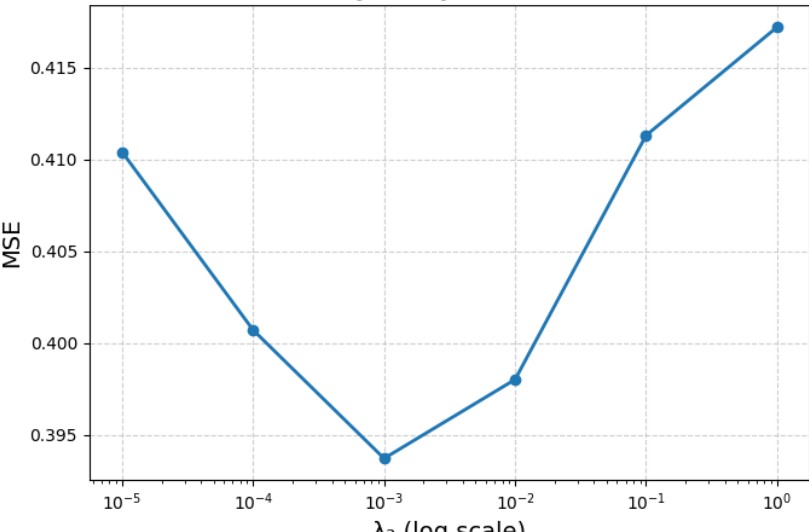

Figure 6: Sensitivity Analysis of $\lambda_2$ on ETTh1 dataset.

## C.4 OPP-MINING RESULTS

OPP-Mining results are summarized in Tables 5 and 6, respectively. In both tables, the first element of each tuple in the second column represents a mined OPP, and the second element denotes the corresponding support of that pattern. It can be observed that as the pattern length increases, the number of frequent OPPs identified at each corresponding length gradually decreases. This reduction in both the number and diversity of frequent OPPs affects the statistical behavior of the constraint term: fewer and more consistent patterns reduce the pairwise covariance among pattern-induced penalties, thereby lowering the variance (and gradient noise) of the constraint loss. We offer the following analysis: For a window length $v$, the mined frequent OPP set is denoted by $F_v$, with normalized weights **slist**(refer to 3.4.3), We let $w_i$ denote **slist**$[i]$. For each pattern $i$, the sample-level constraint penalty is a random variable $g_i$ with mean $\mu_i = \mathbb{E}[g_i]$ and variance $\sigma_i^2 = \mathrm{Var}(g_i)$. The expected constraint loss is:

$$\mathbb{E}[\mathcal{L}_{\text{constraint}}] = \sum_i w_i \mu_i, \tag{20}$$

and its variance decomposes as:

$$\mathrm{Var}(\mathcal{L}_{\text{constraint}}) = \sum_i w_i^2 \sigma_i^2 + 2 \sum_{i<j} w_i w_j \, \mathrm{Cov}(g_i, g_j). \tag{21}$$

The second term captures the correlation between penalties induced by different patterns. When $|F_v|$ is large and many patterns impose conflicting or overlapping requirements, $\mathrm{Cov}(g_i, g_j)$ becomes non-negligible; its accumulation dominates the total variance and turns the constraint term into a source of high training noise. In gradient form, $\mathbf{G} = \sum_i \nabla_\theta(w_i g_i)$, the covariance:

$$\mathrm{Cov}(\mathbf{G}) = \sum_i \mathrm{Cov}(\mathbf{g}_i) + 2\sum_{i<j} \mathrm{Cov}(\mathbf{g}_i, \mathbf{g}_j), \tag{22}$$

is similarly amplified by these pairwise correlations, reducing the gradient signal-to-noise ratio:

$$\mathrm{SNR} = \frac{\|\mathbb{E}[\mathbf{G}]\|}{\sqrt{\mathrm{tr}(\mathrm{Cov}(\mathbf{G}))}}. \tag{23}$$

As $v$ increases to the 'perturbation boundary', the frequent patterns become sparser and more consistent, causing the pairwise covariance terms to drop sharply, the SNR to rise, and the constraint to shift from a source of disturbance to a beneficial prior.

Table 5: Frequent OPPs on the SWaT dataset.

| Pattern length | Mined OPPs and Support counter |
|---|---|
| 8 | $(1,2,3,4,5,6,7,8), 6218$, $(1,2,3,4,5,6,8,7), 2040$, $(1,2,3,4,5,7,8,6), 1283$, $(1,2,3,4,6,7,8,5), 664$, $(1,2,3,4,5,7,6,8), 1029$, $(1,2,3,4,5,8,6,7), 451$, $(1,2,3,4,5,8,7,6), 919$, $(1,2,3,4,6,8,7,5), 701$, $(1,2,3,4,7,8,6,5), 687$, $(1,2,3,5,7,8,6,4), 474$, $(1,2,3,4,6,5,7,8), 989$, $(1,2,3,4,8,7,6,5), 541$, $(1,2,3,5,4,6,7,8), 956$, $(4,5,7,8,6,3,2,1), 460$, $(4,6,7,8,5,3,2,1), 466$, $(5,6,7,8,4,3,2,1), 638$, $(1,2,4,3,5,6,7,8), 979$, $(4,6,8,7,5,3,2,1), 485$, $(5,6,8,7,4,3,2,1), 701$, $(5,7,8,6,4,3,2,1), 710$, $(6,7,8,5,4,3,1,2), 481$, $(6,7,8,5,4,3,2,1), 1104$, $(1,3,2,4,5,6,7,8), 946$, $(5,8,7,6,4,3,2,1), 614$, $(6,8,7,5,4,3,1,2), 499$, $(6,8,7,5,4,3,2,1), 1234$, $(2,3,1,4,5,6,7,8), 514$, $(7,8,6,5,4,2,1,3), 504$, $(7,8,6,5,4,3,1,2), 834$, $(7,8,6,5,4,3,2,1), 2290$, $(2,1,3,4,5,6,7,8), 2192$, $(2,1,3,4,5,6,8,7), 810$, $(2,1,3,4,5,7,8,6), 514$, $(2,1,3,4,5,8,7,6), 453$, $(7,6,8,5,4,3,2,1), 510$, $(3,1,2,4,5,6,7,8), 1242$, $(3,1,2,4,5,6,8,7), 472$, $(4,1,2,3,5,6,7,8), 577$, $(3,2,1,4,5,6,7,8), 1121$, $(3,2,1,4,5,6,8,7), 490$, $(4,2,1,3,5,6,7,8), 785$, $(4,3,1,2,5,6,7,8), 681$, $(5,3,1,2,4,6,7,8), 462$, $(8,7,5,6,4,3,2,1), 1096$, $(4,3,2,1,5,6,7,8), 692$, $(5,4,2,1,3,6,7,8), 477$, $(8,7,6,4,5,3,2,1), 1009$, $(8,7,6,5,1,2,3,4), 602$, $(8,7,6,5,3,4,2,1), 1061$, $(8,7,6,4,2,1,3,5), 496$, $(8,7,6,5,2,1,3,4), 704$, $(8,7,6,5,3,1,2,4), 691$, $(8,7,6,5,4,1,2,3), 1033$, $(8,7,6,5,4,1,3,2), 520$, $(8,7,6,5,4,2,3,1), 999$, $(8,7,6,5,3,2,1,4), 720$, $(8,7,6,5,4,2,1,3), 1228$, $(8,7,6,5,4,3,1,2), 2110$, $(8,7,6,5,4,3,2,1), 6269$ |
| 9 | $(1,2,3,4,5,6,7,8,9), 3570$, $(1,2,3,4,5,6,7,9,8), 1109$, $(1,2,3,4,5,6,8,9,7), 680$, $(1,2,3,4,5,6,8,7,9), 534$, $(1,2,3,4,5,6,9,8,7), 499$, $(1,2,3,4,5,7,6,8,9), 512$, $(1,2,3,4,6,5,7,8,9), 521$, $(1,2,3,5,4,6,7,8,9), 545$, $(1,2,4,3,5,6,7,8,9), 489$, $(7,8,9,6,5,4,3,2,1), 596$, $(1,3,2,4,5,6,7,8,9), 522$, $(7,9,8,6,5,4,3,2,1), 665$, $(8,9,7,6,5,4,3,2,1), 1245$, $(2,1,3,4,5,6,7,8,9), 1196$, $(3,1,2,4,5,6,7,8,9), 655$, $(9,7,8,6,5,4,3,2,1), 576$, $(3,2,1,4,5,6,7,8,9), 580$, $(9,8,6,7,5,4,3,2,1), 588$, $(9,8,7,5,6,4,3,2,1), 595$, $(9,8,7,6,4,5,3,2,1), 564$, $(9,8,7,6,5,3,4,2,1), 523$, $(9,8,7,6,5,4,1,2,3), 545$, $(9,8,7,6,5,4,2,3,1), 512$, $(9,8,7,6,5,4,2,1,3), 662$, $(9,8,7,6,5,4,3,1,2), 1141$, $(9,8,7,6,5,4,3,2,1), 3542$ |
| 10 | $(1,2,3,4,5,6,7,8,9,10), 2127$, $(1,2,3,4,5,6,7,8,10,9), 609$, $(9,10,8,7,6,5,4,3,2,1), 686$, $(2,1,3,4,5,6,7,8,9,10), 641$, $(10,9,8,7,6,5,4,3,1,2), 644$, $(10,9,8,7,6,5,4,3,2,1), 2030$ |
| 11 | $(1,2,3,4,5,6,7,8,9,10,11), 1313$, $(11,10,9,8,7,6,5,4,3,2,1), 1183$ |
| 12 | $(1,2,3,4,5,6,7,8,9,10,11,12), 842$, $(12,11,10,9,8,7,6,5,4,3,2,1), 722$ |
| 13 | $(1,2,3,4,5,6,7,8,9,10,11,12,13), 563$ |

Table 6: Frequent OPPs on the ETTh1 dataset.

| Pattern length | Mined OPPs and Support counter |
|---|---|
| 6 | $(1,2,3,4,5,6), 347$, $(1,2,3,4,6,5), 254$, $(1,2,3,5,6,4), 117$, $(1,2,4,5,6,3), 59$, $(1,2,3,5,4,6), 115$, $(1,2,3,6,4,5), 110$, $(1,2,3,6,5,4), 157$, $(1,2,4,6,5,3), 92$, $(1,2,5,6,4,3), 53$, $(1,2,4,3,5,6), 124$, $(1,2,4,3,6,5), 79$, $(1,2,5,3,4,6), 56$, $(1,2,6,5,3,4), 62$, $(1,2,6,5,4,3), 81$, $(1,3,4,2,5,6), 52$, $(3,5,6,4,2,1), 80$, $(4,5,6,1,2,3), 50$, $(4,5,6,3,1,2), 57$, $(4,5,6,3,2,1), 100$, $(1,3,2,4,5,6), 136$, $(1,3,2,4,6,5), 68$, $(1,3,2,5,4,6), 59$, $(3,6,5,4,2,1), 55$, $(4,6,5,1,2,3), 57$, $(4,6,5,3,1,2), 59$, $(4,6,5,3,2,1), 120$, $(2,3,1,4,5,6), 85$, $(5,6,3,4,2,1), 72$, $(3,4,2,1,5,6), 53$, $(5,6,4,1,2,3), 86$, $(5,6,4,2,3,1), 88$, $(5,6,4,2,1,3), 59$, $(5,6,4,3,1,2), 88$, $(5,6,4,3,2,1), 239$, $(2,1,3,4,5,6), 236$, $(2,1,3,4,6,5), 152$, $(2,1,3,5,6,4), 69$, $(2,1,3,6,5,4), 58$, $(2,1,4,3,5,6), 68$, $(5,4,6,3,1,2), 59$, $(5,4,6,3,2,1), 82$, $(3,1,2,4,5,6), 127$, $(3,1,2,4,6,5), 65$, $(6,3,4,5,1,2), 68$, $(6,3,5,4,1,2), 71$, $(6,4,5,1,2,3), 52$, $(6,4,5,3,1,2), 96$, $(6,4,5,3,2,1), 119$, $(3,2,1,4,5,6), 161$, $(3,2,1,4,6,5), 62$, $(4,2,1,3,5,6), 76$, $(4,3,1,2,5,6), 52$, $(4,3,1,2,5,6), 64$, $(6,5,1,2,3,4), 55$, $(6,5,1,2,4,3), 63$, $(6,5,1,3,4,2), 54$, $(6,5,2,3,4,1), 74$, $(6,5,2,4,3,1), 77$, $(6,5,3,4,1,2), 88$, $(6,5,3,4,2,1), 138$, $(4,3,2,1,5,6), 82$, $(6,5,2,1,3,4), 53$, $(6,5,3,2,4,1), 66$, $(6,5,3,1,2,4), 62$, $(6,5,4,1,2,3), 115$, $(6,5,4,1,3,2), 71$, $(6,5,4,2,3,1), 137$, $(6,5,3,2,1,4), 50$, $(6,5,4,2,1,3), 112$, $(6,5,4,3,1,2), 233$, $(6,5,4,3,2,1), 485$ |
| 7 | $(1,2,3,4,5,6,7), 122$, $(1,2,3,4,5,7,6), 125$, $(1,2,3,4,6,5,7), 50$, $(1,2,3,4,7,6,5), 75$, $(1,2,3,5,7,6,4), 63$, $(4,6,7,5,3,2,1), 52$, $(5,6,7,4,3,2,1), 58$, $(1,3,2,4,5,6,7), 62$, $(5,7,6,4,3,2,1), 65$, $(6,7,5,4,3,2,1), 133$, $(2,1,3,4,5,6,7), 102$, $(2,1,3,4,5,7,6), 70$, $(2,1,3,4,7,6,5), 51$, $(3,1,2,4,5,6,7), 55$, $(7,5,6,4,3,2,1), 57$, $(3,2,1,4,5,6,7), 71$, $(7,6,4,5,3,1,2), 53$, $(7,6,4,5,3,2,1), 53$, $(4,3,2,1,5,6,7), 51$, $(7,6,5,3,4,2,1), 56$, $(7,6,5,4,1,2,3), 67$, $(7,6,5,4,2,1,3), 59$, $(7,6,5,4,3,1,2), 132$, $(7,6,5,4,3,2,1), 234$ |
| 8 | $(7,8,6,5,4,3,2,1), 62$, $(8,7,6,5,4,3,1,2), 64$, $(8,7,6,5,4,3,2,1), 114$ |
| 9 | $(9,8,7,6,5,4,3,2,1), 56$ |

## C.5 EVOLUTION OF OPPCL'S LOSS FUNCTION DURING TRAINING

We present the loss-curve of OPPCL on two of the datasets. Figure 7 illustrates the evolution of the loss function during the training of OPPCL on SWaT dataset and ETTh1 dataset. From parts (a) and (c), it can be observed that the MSE metric of OPPCL—both for non-OPP samples and frequent OPP samples—drops rapidly within the first 10 epochs and then decreases more slowly and stabilizes over the remaining 140 epochs. From parts (b) and (d), we can also see that the constraint loss, which plays a key role in reducing the MSE for frequent OPP samples during training, exhibits a similar trend. This phenomenon suggests that the constraint loss plays a dominant role in guiding the training process, enabling OPPCL to achieve higher predictive accuracy on frequent OPP samples compared with non-OPP samples. This validates the critical role of the pattern constraint loss function in improving the model's performance throughout training.

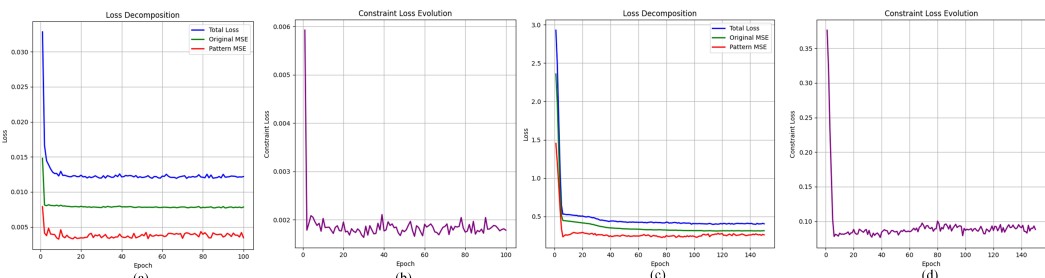

Figure 7: Changes in the loss function during the training of OPPCL on the two datasets. Subfigures (a) and (b) correspond to the SWaT dataset, while (c) and (d) correspond to the ETTh1 dataset.

## C.6 ANALYSIS OF PREDICTION RESULTS FOR RANDOMLY SAMPLED DISCRETE POINTS(ETTH1)

The prediction results for the ETTh1 dataset are presented in Figures 8. As shown in Figure 8, for the ETTh1 dataset, the OPPCL yields a prediction that deviates from the ground truth by only $0.004$ on the first sampled instance from the frequent OPPs set—substantially lower than the deviation produced by the model without contrastive learning (w/o CL), which reaches $0.142$ (i.e., $0.004 \ll 0.142$). Similar results are observed in the second and third sampled instances, where the OPPCL predictions consistently exhibit smaller errors compared with w/o CL. Furthermore, as summarized in Table 2, the overall number of prediction points with smaller absolute errors under OPPCL exceeds that of w/o CL, which is reflected quantitatively in the average error values: $0.3937 < 0.4495$.

## C.7 PREDICTION ANALYSIS ON CONSECUTIVE SAMPLES MATCHING PATTERNS

To further analyze the differences in overall trend prediction, prediction latency, and turning point capture between OPPCL and w/o CL on consecutively matching pattern samples, we extend our evaluation beyond the previous random sampling results. Specifically, we conduct a performance assessment on 1000 consecutive test samples from the SWaT dataset that conform to mined patterns. The comparison between the predicted values and ground truth for both OPPCL and w/o CL is illustrated in Figure 9(a), from which two key conclusions can be drawn:

1). **Trend and Turning Point Alignment.** For OPPCL, the predicted curve closely follows the ground truth in terms of overall trend, with almost no prediction delay. The turning points of both curves are nearly aligned, indicating that OPPCL effectively utilizes prior knowledge to fit the real value trajectory.

2). **Baseline Deviation.** In contrast, w/o CL shows noticeable deviations from the ground truth. There are discrepancies at several turning points, and the prediction values corresponding to those

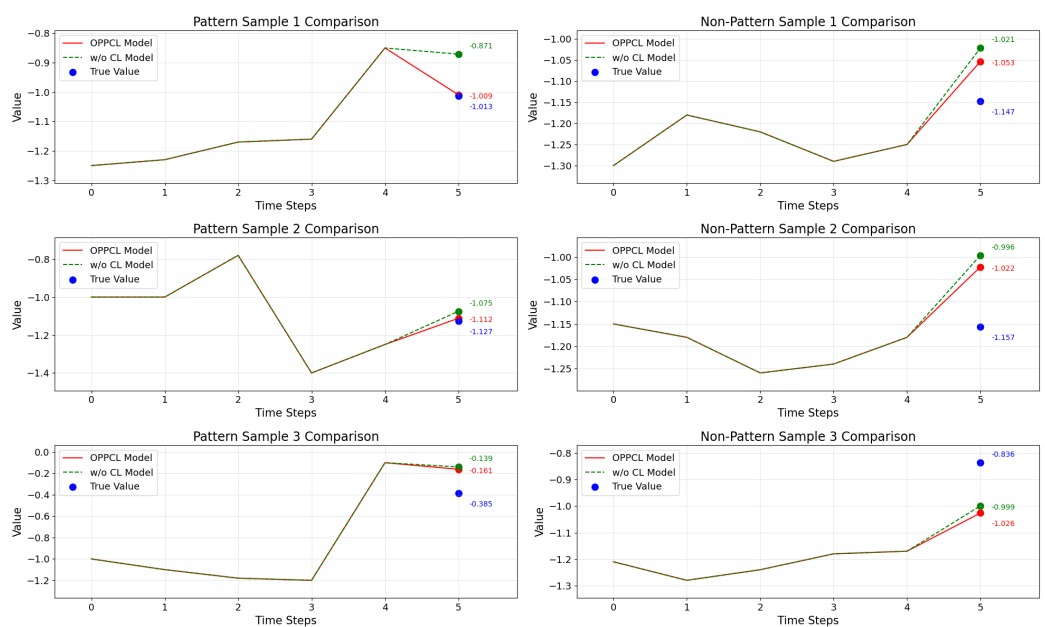

Figure 8: Prediction comparison on three randomly sampled instances from two sets in the ETTh1 dataset.

points diverge more significantly from the actual values. This demonstrates that w/o CL struggles to accurately model complex trends at the ends of pattern-conforming windows.

To more intuitively illustrate the results, Figure 9(b) presents a comparison of the residual distributions between the two models. As shown in the figure, the residuals of OPPCL are significantly more concentrated, with a sharper and narrower peak, indicating that its prediction errors are more tightly centered around zero and thus more accurate. In contrast, w/o CL exhibits a wider and more dispersed tail, with noticeable density beyond the $\pm 0.2$ range. The OPPCL, on the other hand, shows a shorter and steeper tail, suggesting fewer extreme prediction errors and stronger robustness. We further investigate individual prediction errors between the predicted and true values. The re-

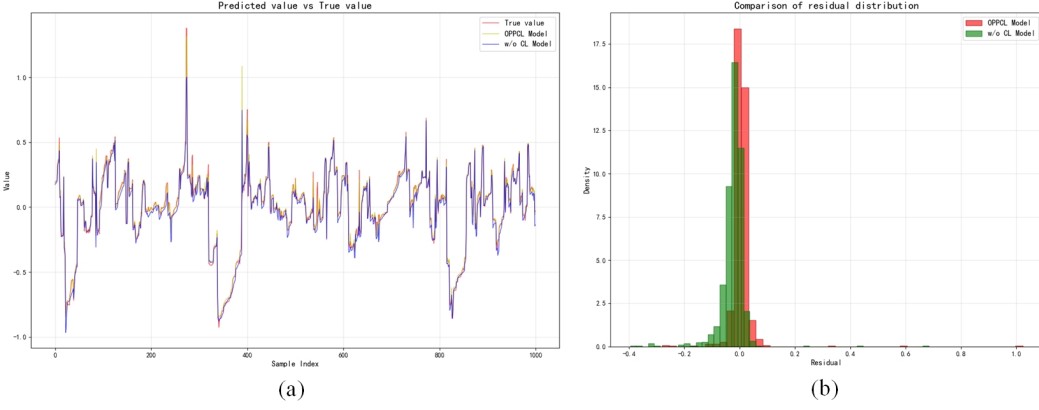

Figure 9: Comparison of prediction results between OPPCL and w/o CL on 1000 consecutive pattern-conforming test samples. Subfigure (a) presents a point-wise comparison of consecutive predictions, whereas subfigure (b) shows their residual distribution.

sults are visualized in Figure 10, which presents a point-to-point comparison between predictions and ground truth over the same 1000 consecutive test samples. From Figure 10, we highlight the following findings:

1). **High Individual Accuracy for OPPCL.** The predictions made by OPPCL are tightly clustered around the ideal prediction line $y = x$, with only a few outliers. This suggests that the pattern constrained loss function effectively enhances the model's ability to learn representations tailored to pattern-conforming samples, resulting in more precise predictions at the individual level.

2). **Greater Deviation in w/o CL.** Without the guidance of the pattern constrained loss, the predictions of the w/o CL model exhibit greater dispersion, particularly at both ends of the value range. This reflects a general tendency toward higher prediction error and less robustness in modeling the structural characteristics of sequential patterns.

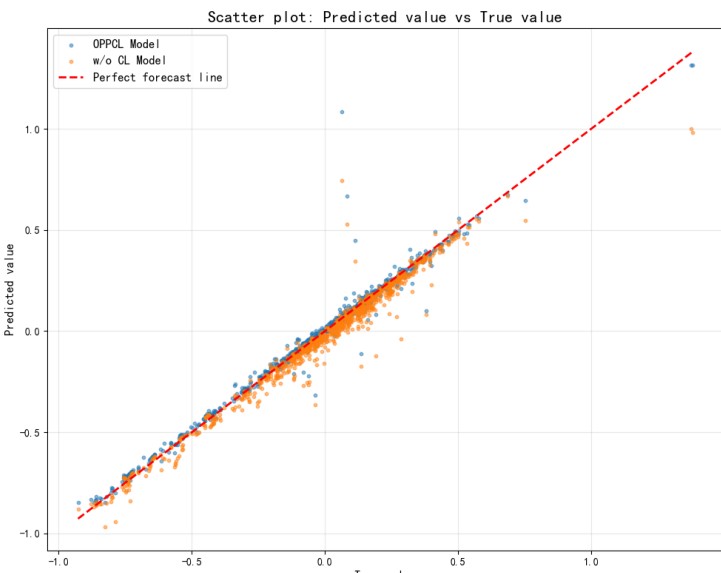

Figure 10: Point-to-point comparison between predicted values and ground truth for 1000 consecutive pattern-conforming test samples by OPPCL and w/o CL.

### C.8 COMPARATIVE ANALYSIS OF ATTENTION ENTROPY(ETTH1)

For the ETTh1 dataset, the distribution and average value comparisons of attention entropy for OPPCL and w/o CL are shown in Figure 11, as illustrated in Figures 11(a) and 11(b). The average entropy for OPPCL ($\mu = 0.348$) is significantly lower than that of w/o CL ($\mu = 0.624$). Furthermore, the bar charts in Figures 11(c) and 11(d) show that, under OPPCL, the $y$-axis values of frequent OPP samples are lower, indicating more concentrated attention.

### C.9 THE QUANTILE-BASED DISTRIBUTIONS OF ATTENTION ENTROPY FOR OPPCL AND W/O CL.

To assess whether this enhanced attention concentration is consistent across the distribution, rather than merely reflected in a lower mean entropy, we further compared the quantile-based distributions of attention entropy for both models. Figure 12 presents the quantile comparison on both datasets.

In Figures 12(a) and 12(b), which show results on the SWaT dataset, the attention entropy values of w/o CL for frequent OPP samples are consistently higher than those of OPPCL in the Q2 and Q3 quantile regions (e.g., $1.276 < 1.673$, $1.656 < 1.819$). Notably, in the Q2 region, w/o CL even exhibits higher attention entropy for pattern samples than for non-OPP samples.

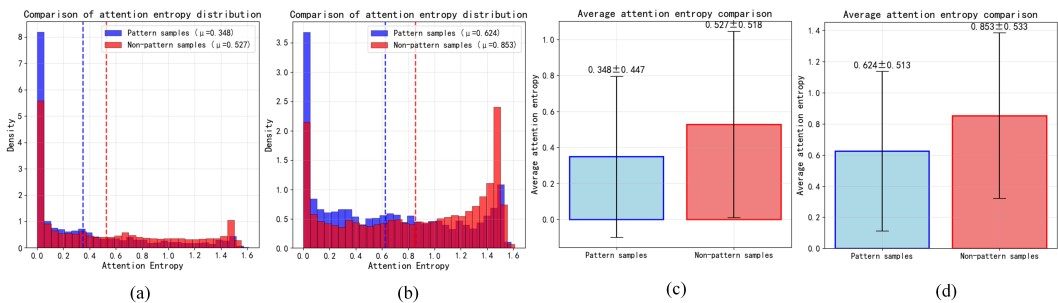

Figure 11: Comparison of attention entropy between OPPCL and w/o CL after training on the ETTh1 dataset, focusing on frequent OPP samples and non-OPP samples. Subfigures (a) and (c) correspond to OPPCL, while (b) and (d) correspond to w/o CL.

Similarly, Figures 12(c) and 12(d) show that on the ETTh1 dataset, the attention entropy values of w/o CL for pattern samples are also higher than those of OPPCL.

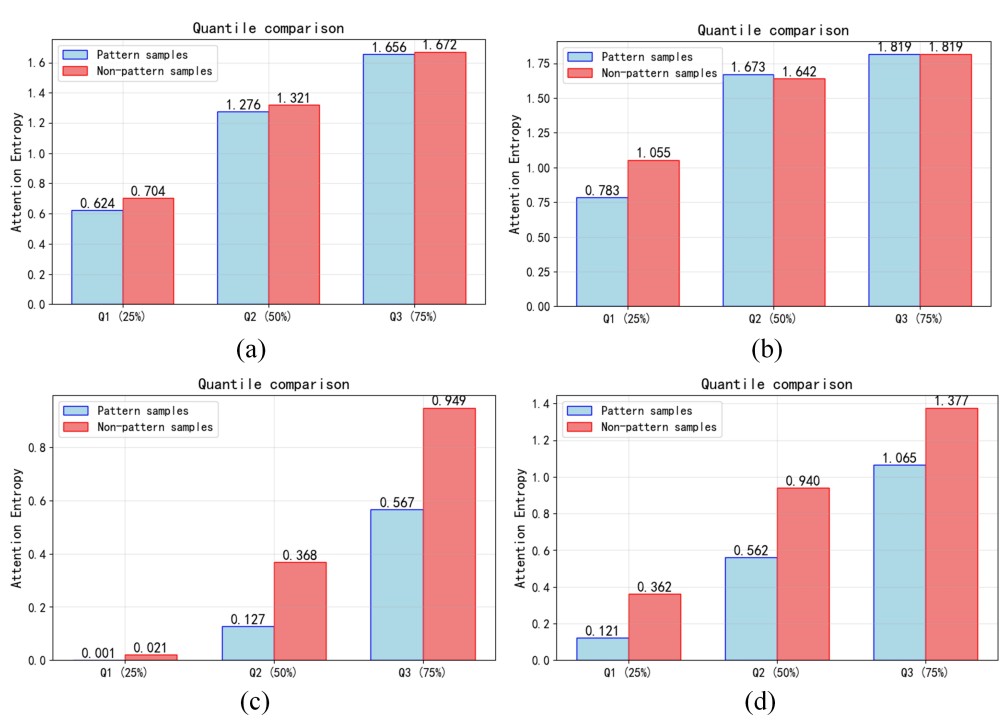

Figure 12: Quantile comparison of attention entropy between OPPCL and w/o CL after training on the SWaT and ETTh1 datasets at the pattern disruption boundary, focusing on frequent OPP samples and non-OPP samples. Subfigures (a) and (b) represent the performance of OPPCL and w/o CL on the SWaT dataset, respectively, while (c) and (d) represent their performance on the ETTh1 dataset.

## C.10 EXPERIMENTAL RESULTS ON LONG-TERM FORECASTING

Table 7 presents the long-term forecasting performance of the models on the SWaT and ETTh1 datasets. For SWaT, OPPCL uses a window size of 9, and for ETTh1, a window size of 7—both corresponding to the 'Perturbation boundary' of the respective datasets. Across 24-, 48-, and 96-

step predictions, OPPCL achieves the best performance on all metrics, demonstrating its ability to maintain steadily increasing errors during recursive long-sequence forecasting. Specifically, for the 24-step prediction, OPPCL attains the lowest MSE of 0.00972, outperforming the second-best TimeCSL(0.01012). Its MAE is also the lowest at 0.05542, indicating more accurate predictions for both local fluctuations and magnitudes. In the 48-step prediction, OPPCL achieves an MSE of 0.01135, 27.2% lower than the second-best LSTNet, with MAE = 0.05901, still the best among all models. Even in the 96-step prediction, a scenario typically prone to error accumulation, OPPCL maintains the lowest MSE and MAE (0.01137 and 0.05907, respectively). Similar results are observed on the ETTh1 dataset.

Overall, OPPCL demonstrates strong long-term forecasting stability, suggesting that the sequentially preserved local pattern constraints effectively mitigate drift errors in recursive predictions.

Table 7: Comparative Evaluation of Models across Prediction Horizons

| Model | SWaT | | | | | | ETTh1 | | | | | |
|---|---|---|---|---|---|---|---|---|---|---|---|---|
| Horizon | 24 | | 48 | | 96 | | 24 | | 48 | | 96 | |
| Metric | MSE | MAE | MSE | MAE | MSE | MAE | MSE | MAE | MSE | MAE | MSE | MAE |
| Autoformer | 1.502e-2 | 7.599e-2 | 3.873e-2 | 1.255e-1 | 2.052e-1 | 2.466e-1 | 2.225e-0 | 1.150e-0 | 2.191e-0 | 1.156e-0 | 2.215e-0 | 1.171e-0 |
| ConvLSTM | 1.139e-2 | 6.265e-2 | 1.955e-2 | 8.738e-2 | 3.499e-2 | 1226e-1 | 2.139e-0 | 1.135e-0 | 2.146e-0 | 1.156e-0 | 2.244e-0 | 1.191e-0 |
| Crossformer | 1.769e-2 | 9.241e-2 | 3.897e-2 | 1.458e-1 | 9.037e-2 | 2.299e-1 | 2.032e-0 | 1.115e-0 | 2.035e-0 | 1.131e-0 | 2.099e-0 | 1.156e-0 |
| Informer | 3.305e-2 | 1346e-1 | 7.158e-2 | 1.999e-1 | 1.385e-1 | 2.779e-1 | 2.051e-0 | 1.122e-0 | 2.114e-0 | 1.153e-0 | 2.215e-0 | 1.187e-0 |
| LSTNet | 1.027e-2 | 5.542e-2 | 1.558e-2 | 7.399e-2 | 2.673e-2 | 1.032e-1 | 2.156e-0 | 1.134e-0 | 2.137e-0 | 1.147e-0 | 2.193e-0 | 1.173e-0 |
| MegaCRN | 1.622e-2 | 8.571e-2 | 3.415e-2 | 1.324e-1 | 7.205e-2 | 2.000e-1 | 2.348e-0 | 1.197e-0 | 2.651e-0 | 1.293e-0 | 3.016e-0 | 1.393e-0 |
| SCINet | 3.175e-2 | 1.166e-1 | 5.793e-2 | 1.566e-1 | 8.400e-2 | 1.871e-1 | 2.000e-0 | 1.096e-0 | 2.065e-0 | 1.140e-0 | 2.181e-0 | 1.178e-0 |
| Leddam | 1.089e-2 | 6.256e-2 | 1.929e-2 | 9.126e-2 | 3.853e-2 | 1.396e-1 | 2.006e-0 | 1.108e-0 | 2.018e-0 | 1.125e-0 | 2.064e-0 | 1.146e-0 |
| TimeCSL | 1.012e-2 | 5.498e-2 | 1.631e-2 | 7.408e-2 | 2.899e-2 | 1.039e-1 | 2.062e-0 | 1.121e-0 | 2.026e-0 | 1.127e-0 | 2.058e-0 | 1.145e-0 |
| CycleNet | 1.097e-2 | 6.016e-2 | 1.900e-2 | 8.389e-2 | 3.372e-2 | 1.184e-1 | 2.114e-0 | 1.135e-0 | 2.200e-0 | 1.173e-0 | 2.296e-0 | 1.208e-0 |
| **OPPCL** | **9.720e-3** | **5.469e-2** | **1.135e-2** | **5.901e-2** | **1.137e-2** | **5.907e-2** | **1.948e-0** | **1.096e-0** | **2.015e-0** | **1.120e-0** | **2.026e-0** | **1.133e-0** |

