# OpenReview forum: "Order-Preserving Pattern Mining Enhances Structure-Aware Time Series Forecasting"
_ICLR.cc/2026/Conference — ICLR 2026 Conference Withdrawn Submission_

### Official Review · Reviewer_guBk · 2025-10-17

**Soundness:** 2
**Presentation:** 1
**Contribution:** 2
**Rating:** 2
**Confidence:** 4

**Summary:**

This paper introduces Order-Preserving Patterns (OPPs) into time series forecasting and proposes the OPPCL (Order-Preserving Pattern Constrained Learning) model. The model employs a CNN for dimensionality reduction of high-dimensional labeled time series, uses a sliding window and support counting strategy to mine frequent OPPs, designs an OPP matching mechanism to distinguish between OPP and non-OPP training samples, and develops a pattern-constrained loss function (incorporating right-boundary, left-boundary, and intermediate position constraints) to align predictions with prior pattern logic.

**Strengths:**

1.This paper propose a pattern-aware forecasting model OPPCL, which explicitly introduces frequent OPPs as structural priors in the time series forecasting task, this paper design a position-sensitive pattern-constrained loss function, which explicitly supervises the relative order of the model outputs；
2.The method proposed in the paper has proven its effectiveness on SWAT and ETTh1 datasets；
3.Most parts of the paper are comprehensible.

**Weaknesses:**

1.The method proposed in the paper is only tested on two datasets. Current mainstream time series research typically validates models on more than 10 datasets, so testing on merely two datasets is far from sufficient;
2.Using only MSE as the evaluation metric is also insufficient. In Table 2, the optimal results under certain settings are not highlighted in bold;
3.The figure quality of the paper needs improvement: the text in many figures is too small to read clearly; Figure 1 lacks strong persuasiveness; much text in Figure 2 is outside the text boxes; many numerical labels in Figure 3 are also outside the graph, and the x-axis and y-axis labels are not marked;
4.The paper contains an excessive number of formulas. Many general formulas (such as the calculation of attention) do not need to be listed, and many formulas lack punctuation marks;
5.The dimensionality reduction strategy lacks consistency and detailed validation. The paper uses CNN for SWaT (51-dimensional data) and PCA for ETTh1 (7-dimensional data) but does not justify this choice (e.g., why not use CNN for ETTh1 or PCA for SWaT?);
6.The pattern constraint loss function’s hyperparameter setting lacks sensitivity analysis. The total loss uses identical weights for both datasets, but the paper does not explore how varying these hyperparameters influences model performance;
7.Please explain the sentence in the Conclusion section: "Furthermore, OPPCL enhances the attention focus of baseline Transformer-based models, indicating improved interpretability";
8.The references and notation consistency require refinement. Some references lack complete publication details (e.g., "Wang & Chen" for LSTNet is not fully cited in the References section).

**Questions:**

Please see Weaknesses.

---

> ### Author Response · Authors · 2025-11-27
>
> We sincerely thank you for your valuable review. Your comments have greatly helped improve our manuscript. In response to your suggestions, we have made the following revisions:
>
> 1. Expanded datasets: We have added eight public datasets to comprehensively evaluate OPPCL, including ETTh2, ETTm1, ETTm2, PowerSystem, Weather, Electricity, Exchange_Rate, and Traffic. Please refer to Section 4.3 for details.
>
> 2. MAE metric: We have included the MAE metric and conducted new experiments for all models on each dataset. Detailed results are provided in Section 4.3.
>
> 3. Figures refinement: Figures 1, 3, 4, 8, 9, 10, 11, and 12 have been refined, including improvements to legends, font sizes, addition of axis labels in Figures 3 and 8, and ensuring that numbers do not exceed the figure boundaries. Figure 1 also now includes more detailed explanations of pattern trends.
>
> 4. Formula simplification: Some general formulas have been streamlined, including omitting detailed calculations of the attention mechanism. The layout of single-line formulas has been reorganized, keeping only the most essential formulas, and all symbols have been carefully checked. Please see Section 3 for details.
>
> 5. Dimensionality reduction strategy: We have discussed the choice of dimensionality-reduction methods in the manuscript. Among the eight added datasets, PowerSystem and the original SWaT dataset contain per-time-point labels (i.e., anomaly annotations). OPPCL was originally developed for SWaT, so we designed a CNN-based dimensionality-reduction module for labeled time series. ETTh1 is unlabeled, so standard PCA is applied. For the new PowerSystem dataset, we also use CNN-based reduction, and for the remaining seven unlabeled datasets, PCA is retained. Further details, including a table indicating which datasets have labels, can be found in Appendix C.1.
>
> 6. Parameter sensitivity analysis: In response to your comment regarding parameter sensitivity, we first analyzed the selection of the
> $\lambda_2$ parameter and then designed corresponding experiments. We tested
> $\lambda_2$ values of 0.00001, 0.0001, 0.001, 0.01, 0.1, and 1.0, and evaluated their impact on the final MSE. Results are presented in Figures 5 and 6 of Appendix C.3.
>
> 7. Interpretability of conclusions: We provide the following explanation regarding model interpretability. We quantify the concentration of the model’s attention distribution using attention entropy; lower attention entropy indicates that the model focuses more on key input positions rather than uniformly across all time steps, thereby improving interpretability. In Section 4.4.2 and Figure 4, we show that OPPCL achieves significantly lower attention entropy than the unconstrained version (w/o CL). For example, on the SWaT dataset, the average attention entropy of frequent OPP samples decreases from 1.295 (w/o CL) to 1.112 (OPPCL). Lower attention entropy indicates that: (1) the model focuses more on positions relevant to frequent OPPs during prediction; (2) the attention distribution is more stable and consistent; (3) decisions in structured pattern regions are clearer. Because OPPCL’s structured loss enforces the relative order logic of frequent OPPs, it naturally emphasizes attention on key positions that match the OPP, resulting in more concentrated attention, reduced noise, and improved interpretability.
>
> 8. References: We have updated the references by adding available publication information.
>
> Once again, we sincerely thank you for your review and constructive feedback. We wish you all the best!

---

### Official Review · Reviewer_M3pn · 2025-10-31

**Soundness:** 3
**Presentation:** 3
**Contribution:** 3
**Rating:** 6
**Confidence:** 3

**Summary:**

This paper introduces OPPCL, a time series forecasting model designed to be "structure-aware" by explicitly incorporating Order-Preserving Patterns (OPPs). The core idea is to mine frequent relative orderings from the training data and then use a novel pattern-constrained loss function to penalize predictions that violate these learned structures.

**Strengths:**

The main strength of this work lies in its conceptual novelty. Moving beyond implicit structure learning to an explicit, pattern-based constraint is a significant and promising direction, especially for industrial time series where underlying processes often dictate such ordinal logic. The proposed constraint loss is an intuitive and direct implementation of this idea. The authors support their claims with a comprehensive set of experiments against numerous baselines, and the ablation study effectively demonstrates the utility of the proposed constraint mechanism.

**Weaknesses:**

However, the paper suffers from several critical weaknesses that currently undermine the validity and generality of its contributions. The most significant issue is a lack of clarity and consistency in the core methodology. The dimensionality reduction step—a crucial prerequisite for pattern mining—is described as a CNN trained on unexplained "timestamp-level labels" for one dataset, while a simple PCA is used for another. This inconsistency feels ad-hoc and makes the framework seem less like a generalizable solution.

The reported performance improvements on the SWaT dataset are so extraordinary (orders of magnitude better than strong baselines) that they raise concerns about the experimental protocol. It is essential to confirm that the OPP mining process was conducted strictly on the training set, with no leakage from the test data. Without this confirmation, the results are difficult to trust. Finally, the "Perturbation Boundary" concept, invoked to explain why the model's advantage is limited to specific input lengths, feels like a post-hoc rationalization for a potential lack of robustness rather than a deeply analyzed phenomenon.

**Questions:**

N/A

---

> ### Author Response · Authors · 2025-11-27
>
> Thank you for your review. Your comments have been invaluable in improving our manuscript. In response to your suggestions, we have made the following revisions:
>
> 1. On the dimensionality-reduction concern. We have addressed this issue in the manuscript. Specifically, we added eight additional datasets. Among them, the PowerSystem dataset and the originally included SWaT dataset contain per-time-point labels (i.e., anomaly annotations). In fact, OPPCL was originally developed with SWaT in mind, so we designed a CNN-based dimensionality-reduction module for labeled time series. The ETTh1 dataset is unlabeled, and therefore we adopt the standard PCA-based dimensionality reduction for it. For the newly added PowerSystem dataset we also apply our CNN-based reduction. For the remaining seven unlabeled datasets, we retain PCA. A more detailed description (including a table indicating which datasets have labels) is provided in Appendix C.1. Thank you again for this comment.
>
> 2. On the mining procedure and data leakage. We understand your concern regarding the mining process. We ensure that all pattern mining is performed strictly on the training split; no information from the test set is used, and there is no data leakage.
>
> 3. On the phenomenon you raised regarding the “perturbation boundary”. We have carried out a more in-depth analysis of this issue(We have added the details to Appendix C.4, and we sincerely invite you to view them.):
>
> For a window length $v$, the mined frequent OPP set is denoted by $F_v$,
> with normalized weights $\mathbf{slist}$ (refer to Sec.~3.4.3).
> We let $w_i$ denote $\mathbf{slist}[i]$.
> For each pattern $i$, the sample-level constraint penalty is a random
> variable $g_i$ with mean $\mu_i = \mathbb{E}[g_i]$ and variance
> $\sigma_i^{2} = \operatorname{Var}(g_i)$.
> The expected constraint loss is:
>
> $\mathbb{E}[\mathcal{L}_{\mathrm{constraint}}] = \sum_i w_i \mu_i.$
>
>
> Its variance decomposes as:
>
> $\mathrm{Var}(L_{\mathrm{constraint}}) = \sum_i w_i^2 \sigma_i^2 + 2 \sum^{i<j} w_i w_j \mathrm{Cov}(g_i, g_j)$
>
> The second term captures the correlation between penalties induced by
> different patterns. When $|F_v|$ is large and many patterns impose
> conflicting or overlapping requirements, $\operatorname{Cov}(g_i, g_j)$
> becomes non-negligible; its accumulation dominates the total variance and
> turns the constraint term into a source of high training noise.In gradient form, $\mathbf{G} = \sum_i \nabla_\theta (w_i g_i)$,
>
> the covariance:
>
> $\mathrm{Cov}(\mathbf{G}) = \sum_i \mathrm{Cov}(\mathbf{g}_i) + 2 \sum^{i<j} \mathrm{Cov}(\mathbf{g}_i, \mathbf{g}_j)$
>
>
>
> is similarly amplified by these pairwise correlations, reducing the gradient signal-to-noise ratio:
>
> $\mathrm{SNR} = \frac{\| \mathbb{E}[\mathbf{G}] \|}{\sqrt{\mathrm{tr}(\mathrm{Cov}(\mathbf{G}))}}$
>
>
> As $v$ increases toward the ''perturbation boundary'', the frequent patterns
> become sparser and more consistent, causing the pairwise covariance terms
> to decrease sharply, the SNR to rise, and the constraint to shift from
> a source of disturbance into a beneficial prior.
>
> We sincerely thank you again for your review.

---

### Official Review · Reviewer_yWsc · 2025-10-31

**Soundness:** 2
**Presentation:** 3
**Contribution:** 2
**Rating:** 2
**Confidence:** 4

**Summary:**

This paper proposes Order-Preserving Patterns (OPPs) for time series forecasting and leverages frequent OPPs as explicit priors. Experiments on 2 datasets show improvements.

**Strengths:**

1. This paper is easy to follow.
2. Visualization and showcases.
3. Code are provided for reproducibility.

**Weaknesses:**

1. Many related works are missing, such as [1-2] which also consider similar Order, Shapelet or Shape for time series modeling. The claimed contributions and novelty are limited.

[1] TimeCSL: Unsupervised Contrastive Learning of General Shapelets for Explorable Time Series Analysis. PVLDB 2024.

[2] Shape analysis for time series. NeurIPS 2024.

[3] Revitalizing multivariate time series forecasting:Learnable decomposition with inter-series dependencies and intra-series variations modeling. ICML 2024.

2. Key experimental comparison for baselines [1-4] is missing.

[4] Deep Time Series Forecasting With Shape and Temporal Criteria.

3. More benchmarks, such as Traffic / Electricity / Weather, and metrics, such as MAE, should be considered. The authors should justify their choice of benchmarks and metrics, and explain why they believe their current selections are sufficient or how they plan to expand their evaluation.

4. The author should also conduct experiments and compare results on long-term forecasting. And the used baselines are out-of-date. Please compare state-of-the-art baselines such as [3].

**Questions:**

see weaknesses.

---

> ### Author Response · Authors · 2025-11-26
>
> We would like to sincerely thank you for your thorough and constructive review. Your feedback has been invaluable in improving the quality of our manuscript. In response to your comments, we have made the following revisions: 1)We have expanded the related work section to address the missing discussions you pointed out. 2)Following your suggestion, we have included additional state-of-the-art baseline models for comparison and we have incorporated the MAE metric and added experiments on eight publicly available datasets. 3)Considering the unique characteristics of OPPCL, we have additionally designed long-term forecasting experiments to further validate its effectiveness. I elaborate on these points as follows:
>
> 1.Regarding your comments on the related work section, we kindly invite you to review the updated version of our manuscript. We have revised and expanded the related work section by incorporating the missing components you pointed out.
>
> 2.In response to your concerns regarding the baseline datasets, evaluation metrics, and the need for more advanced comparison models, we have expanded our experiments accordingly. Specifically, we have incorporated eight additional public benchmark datasets, included MAE as an additional evaluation metric, and added three state-of-the-art baseline models—the Leddam model (ICML 2024), the TimeCSL model (PVLDB 2024), and the CycleNet model (NeurIPS 2024). Due to the large amount of tabular information, we kindly invite you to refer directly to Tables 1 and 2 in the revised manuscript. In particular, Table 2 has been newly added to present the MAE results of all models across the ten benchmark datasets.
>
> 3.In addition, we evaluated the long-term forecasting capability of OPPCL. Theoretically, OPPCL specifies a window size based on the mined pattern lengths: all elements in the window except the last one are used as pattern-matching signals, while the prediction of the final element is structurally constrained (see Section~3.4.3 for details). Consequently, the maximum feasible window length is determined by the length of the most frequent OPP obtained from prior knowledge. To conduct long-term forecasting while preserving the structural constraints, we apply the following strategy during testing: OPPCL performs recursive prediction, where at each step the most recent $v - 1$ real or previously predicted values are fed into the model to predict the next point $\hat{y}_{t+1}$. The prediction is then appended to the sequence, and the process is rolled forward until the desired horizon is reached. The long-term forecasting results of OPPCL on the SWaT and ETTh1 datasets are presented in Appendix C.10. We'll give a brief demonstration here:
> ### Comparison of Models (MSE only, three horizons)
>
> | Model      | SWaT-24 | SWaT-48 | SWaT-96 | ETTh1-24 | ETTh1-48 | ETTh1-96 |
> |------------|---------|---------|---------|----------|----------|----------|
> | Autoformer | 1.502e-2 | 3.873e-2 | 2.052e-1 | 2.225e-0 | 2.191e-0 | 2.215e-0 |
> | ConvLSTM   | 1.139e-2 | 1.955e-2 | 3.499e-2 | 2.139e-0 | 2.146e-0 | 2.244e-0 |
> | Crossformer| 1.769e-2 | 3.897e-2 | 9.037e-2 | 2.032e-0 | 2.035e-0 | 2.099e-0 |
> | Informer   | 3.305e-2 | 7.158e-2 | 1.385e-1 | 2.051e-0 | 2.114e-0 | 2.215e-0 |
> | LSTNet     | 1.027e-2 | 1.558e-2 | 2.673e-2 | 2.156e-0 | 2.137e-0 | 2.193e-0 |
> | MegaCRN    | 1.622e-2 | 3.415e-2 | 7.205e-2 | 2.348e-0 | 2.651e-0 | 3.016e-0 |
> | SCINet     | 3.175e-2 | 5.793e-2 | 8.400e-2 | 2.000e-0 | 2.065e-0 | 2.181e-0 |
> | Leddam     | 1.089e-2 | 1.929e-2 | 3.853e-2 | 2.006e-0 | 2.018e-0 | 2.064e-0 |
> | TimeCSL    | 1.012e-2 | 1.631e-2 | 2.899e-2 | 2.062e-0 | 2.026e-0 | 2.058e-0 |
> | CycleNet   | 1.097e-2 | 1.900e-2 | 3.372e-2 | 2.114e-0 | 2.200e-0 | 2.296e-0 |
> | **OPPCL**  | **9.720e-3** | **1.135e-2** | **1.137e-2** | **1.948e-0** | **2.015e-0** | **2.026e-0** |
>
>
> ### Comparison of Models (MAE only, three horizons)
>
> | Model      | SWaT-24 | SWaT-48 | SWaT-96 | ETTh1-24 | ETTh1-48 | ETTh1-96 |
> |------------|---------|---------|---------|----------|----------|----------|
> | Autoformer | 7.599e-2 | 1.255e-1 | 2.466e-1 | 1.150e-0 | 1.156e-0 | 1.171e-0 |
> | ConvLSTM   | 6.265e-2 | 8.738e-2 | 1.226e-1 | 1.135e-0 | 1.156e-0 | 1.191e-0 |
> | Crossformer| 9.241e-2 | 1.458e-1 | 2.299e-1 | 1.115e-0 | 1.131e-0 | 1.156e-0 |
> | Informer   | 1.346e-1 | 1.999e-1 | 2.779e-1 | 1.122e-0 | 1.153e-0 | 1.187e-0 |
> | LSTNet     | 5.542e-2 | 7.399e-2 | 1.032e-1 | 1.134e-0 | 1.147e-0 | 1.173e-0 |
> | MegaCRN    | 8.571e-2 | 1.324e-1 | 2.000e-1 | 1.197e-0 | 1.293e-0 | 1.393e-0 |
> | SCINet     | 1.166e-1 | 1.566e-1 | 1.871e-1 | 1.096e-0 | 1.140e-0 | 1.178e-0 |
> | Leddam     | 6.256e-2 | 9.126e-2 | 1.396e-1 | 1.108e-0 | 1.125e-0 | 1.146e-0 |
> | TimeCSL    | 5.498e-2 | 7.408e-2 | 1.039e-1 | 1.121e-0 | 1.127e-0 | 1.145e-0 |
> | CycleNet   | 6.016e-2 | 8.389e-2 | 1.184e-1 | 1.135e-0 | 1.173e-0 | 1.208e-0 |
> | **OPPCL**  | **5.469e-2** | **5.901e-2** | **5.907e-2** | **1.096e-0** | **1.120e-0** | **1.133e-0** |

---

### Official Review · Reviewer_kqFx · 2025-11-01

**Soundness:** 3
**Presentation:** 1
**Contribution:** 2
**Rating:** 0
**Confidence:** 5

**Summary:**

The authors used an incorrect template (wrong page margin)?

**Strengths:**

N/A

**Weaknesses:**

N/A

**Questions:**

N/A

---

> ### Author Response · Authors · 2025-11-27
>
> We sincerely appreciate your careful review. In this round of revisions, we have thoroughly checked and adjusted the page margins. We have also carefully addressed and refined the manuscript in response to all comments from the other three reviewers. If you have time, we would be grateful if you could kindly review the updated version of our manuscript. We sincerely apologize for any inconvenience this may have caused.

---

### Note · Authors · 2026-01-12

I have read and agree with the venue's withdrawal policy on behalf of myself and my co-authors.